# Generative Visual Prompt: Unifying Distributional Control of Pre-Trained Generative Models

**Chen Henry Wu,   Saman Motamed,   Shaunak Srivastava,   Fernando De la Torre**

Robotics Institute, Carnegie Mellon University, Pittsburgh, PA

{chenwu2,ftorre}@cs.cmu.edu, {saman.moatamed,shaunak1999}@gmail.com

## Abstract

Generative models (e.g., GANs, diffusion models) learn the underlying data distribution in an unsupervised manner. However, many applications of interest require sampling from a particular region of the output space or sampling evenly over a range of characteristics. For efficient sampling in these scenarios, we propose **Generative Visual Prompt** (PromptGen), a framework for distributional control over pre-trained generative models by incorporating knowledge of other off-the-shelf models. PromptGen defines control as energy-based models (EBMs) and samples images in a feed-forward manner by approximating the EBM with invertible neural networks, avoiding optimization at inference. Our experiments demonstrate how PromptGen can efficiently sample from several unconditional generative models (e.g., StyleGAN2, StyleNeRF, diffusion autoencoder, NVAE) in a controlled or/and de-biased manner using various off-the-shelf models: (1) with the CLIP model as control, PromptGen can sample images guided by text, (2) with image classifiers as control, PromptGen can de-bias generative models across a set of attributes or attribute combinations, and (3) with inverse graphics models as control, PromptGen can sample images of the same identity in different poses. (4) Finally, PromptGen reveals that the CLIP model shows a "reporting bias" when used as control, and PromptGen can further de-bias this controlled distribution in an iterative manner.[1]

## 1   Introduction

Generative models learn the underlying high-dimensional data distribution and have achieved promising performance on image synthesis [4, 75, 36, 21]. Though being well praised, they still face two main limitations. First, since generative models are typically trained in an unsupervised way, they lack controllability, meaning that it is unclear how to sample from a specific region of the space. Second, generative models are prone to inherit the imbalance and bias of training data [60, 31]. For instance, StyleGAN2 is more likely to produce images of white individuals, see Figure 1(e). Previous works have studied these challenges separately, and typical methods include editing of "style" codes [23, 68, 31] and explicit conditions [43, 69]. However, these methods are either model-dependent (i.e., requiring a well-structured style space) or label-intensive (i.e., requiring all training samples to be labeled for explicit conditions), limiting their generality and practical use.

To address the above challenges, this paper advocates a unified formulation, *distributional control of generative models*, which enables controllability by incorporating the knowledge of off-the-shelf models (e.g., CLIP [58], classifiers, or inverse graphics models [16]). Based on this unified view, we propose to learn distributions in the latent space of a pre-trained generative model while keeping the pre-trained weights fixed. Given its conceptual similarity to prompting [42, 85, 86, 30, 22], we term our framework as Generative Visual Prompt (PromptGen). PromptGen requires **no training data**, and the only supervision comes from off-the-shelf models that help define the control. Specifically,

---

[1]The code is available at https://github.com/ChenWu98/Generative-Visual-Prompt.

36th Conference on Neural Information Processing Systems (NeurIPS 2022).

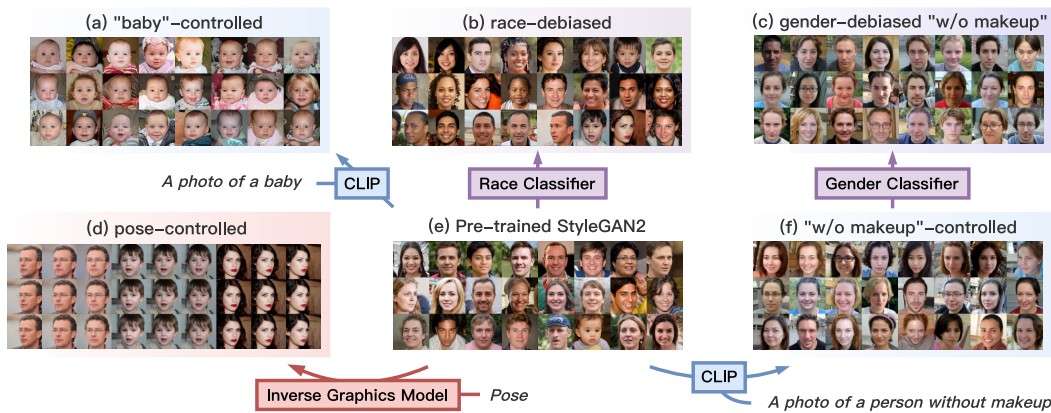

Figure 1: PromptGen uses different off-the-shelf models (e.g., CLIP [58], inverse graphics models, and classifiers) to control the output distribution of a pre-trained and fixed generative model (e.g., StyleGAN2). Colored boxes (e.g., the blue CLIP) indicate the control. See the text for details.

PromptGen allows the user to define controls using an energy-based model (EBM) approximated by an invertible neural network (INN). Unlike methods that require optimization at inference in EBM sampling [49, 14, 45, 52], PromptGen samples images in a **feed-forward** manner, which is highly efficient. Moreover, PromptGen **stands alone** at inference, meaning that we can discard the off-the-shelf models, which define the control, after training.

We illustrate the benefits of PromptGen with several experiments. Figure 1 demonstrates our main findings with StyleGAN2 trained on FFHQ [35] without any labels, while we also show results for StyleNeRF [21], diffusion autoencoder [57], and NVAE [75] in Section 4. Figure 1(a) illustrates how we can sample from StyleGAN2 based on text descriptions such as "a photo of a baby". Figure 1(b) shows that PromptGen can leverage a race classifier (potentially trained on a different dataset) to sample uniformly across all races, de-biasing the pre-trained StyleGAN2. Figure 1(d) illustrates that PromptGen can generate images of the same identity in different poses, guided by a pose regressor.

Finally, it is worth pointing out that PromptGen not only offers **generality** for algorithmic design and **modularity** for control composition, but also enables **iterative** controls when some controls are contingent on others. For instance, one may train a text-controlled distribution and then de-bias this distribution. To achieve this, we view the composition of the INN and the generative model as a new "generative model" to be controlled. Figure 1(f) illustrates that PromptGen reveals "reporting bias" of the CLIP model [58], where "without makeup" is – perhaps surprisingly – positively correlated with female, and Figure 1(c) shows that PromptGen can further mitigate this bias with iterative control.

Table 1: Comparison between methods.

|  | StyleFlow [1] | PPGM [49] / LACE [52] | Guided DDPM [10] | PromptGen |
|---|---|---|---|---|
| Arbitrary control (e.g., CLIP) | ✗ | ✓ | ✓ | ✓ |
| Low-dimensional latent space | ✓ | ✓ | ✗ | ✓ |
| Stands alone at inference | ✓ | ✗ | ✗ | ✓ |
| Feed-forward (i.e., no inference-time optim.) | ✓ | ✗ | ✗ | ✓ |
| Iterative distributional control | ✗ | ✗ | ✗ | ✓ |

## 2    Related Work

Over the past few years, generative models have gained the ability to generate images with high visual quality. A few of the most widely used methods include generative adversarial networks (GANs) [18], VAE [40], invertible neural networks [11, 12], and diffusion models [25, 71]. In particular, generative models trained on large amounts of unlabeled data, e.g., BigGAN [4], StyleGANs [35, 36, 34, 64], Glow [39], and diffusion models [51, 10], achieve promising image synthesis results.

Despite their success, controllability and de-biasing are still two fundamental challenges generative models face. For controllability, existing methods include explicit conditioning at training [43, 69] or local editing of the learned representation, e.g, "style" codes [1, 56, 46, 68, 80]. For de-biasing, existing methods include local editing of "style" codes [60, 31] and importance sampling for either training [20] or inference [29]. Existing works study these problems separately, each requiring a specific design for the task studied. On the contrary, PromptGen is a unified framework for arbitrary controls defined by off-the-shelf models. The benefits of unifying tasks have been shown by a recent trend of works in multiple research areas across vision and language [59, 48, 63, 81, 61]. Moreover, one can fine-tune generative models [17] for domain adaptation, which is orthogonal to PromptGen: since PromptGen maps a generative model to another generative model (Algorithm 1), fine-tuning can be applied before or after PromptGen training. We leave this exploration to future studies.

Previous methods usually sample from EBM [41] with Markov Chain Monte Carlo (MCMC) [73, 76, 14, 19, 13, 45]. Among them, plug-and-play generative models (PPGMs) [49] and LACE [52] define latent-space EBMs. However, MCMC requires inference-time optimization, which is inefficient and requires the off-the-shelf models to be available at inference; this is also the criticism for diffusion models, regardless of being gradient-guided [51, 67] or not [25, 26, 50]. In contrast, PromptGen achieves efficient, feed-forward sampling. Table 1 shows a comparison with previous methods. Our INN training is similar to that proposed by [54] to sample from physical systems, which is later used by [78] to solve inverse problems with two composed INNs. Notably, [54] and [78] do not leverage a low-dimensional latent space, and [78] requires training a separate model for each image sample. In this paper, we use INN to model arbitrary EBMs for various generative models.

Figure 2: Overview of Generative Visual Prompt (PromptGen). Given a pre-trained generative model $G$ and a control $\mathcal{C}$, we learn a distribution $p(z|\mathcal{C})$ in $G$'s latent space, while keeping $G$ fixed. Each control $\mathcal{C}$ can have multiple components, e.g., $\mathcal{C} = \{\text{text} = \text{"a photo of a baby"}, \text{gender} = \texttt{male}\}$. PromptGen views the composition $G \circ f_{\boldsymbol{\theta}}$ as a new "generative model" for iterative control.

## 3   Method

Figure 2 illustrates our PromptGen framework. To begin, the user selects a pre-trained generative model $G$. PromptGen then lets the user specify a control $\mathcal{C}$ as an energy-based model (EBM) $p(z|\mathcal{C})$. We train an INN $f_{\boldsymbol{\theta}}$ to approximate $p(z|\mathcal{C})$. PromptGen views the functional composition $G \circ f_{\boldsymbol{\theta}}$ as a new generative model and can perform iterative control. Algorithm 1 describes the overall procedure.

---

**Algorithm 1:** Generative Visual Prompt (PromptGen)

---

**Input:** Generative model $G : \mathbb{R}^d \to \mathcal{X}$
**repeat**
  1. **Input:** control $\mathcal{C}$ of the current iteration
  2. Define an EBM $p(z|\mathcal{C})$ for $\mathcal{C}$   (Section 3.1)
  3. Train an INN $f_{\boldsymbol{\theta}} : \mathbb{R}^d \to \mathbb{R}^d$ to approximate $p(z|\mathcal{C})$   (Section 3.2)
  4. $G \leftarrow G \circ f_{\boldsymbol{\theta}}$
**until** *user stops the iteration*
**return** $G : \mathbb{R}^d \to \mathcal{X}$, which is a feed-forward network

---

### 3.1   Latent-Space EBM for Distributional Control

The plug-and-play generative model [49] was first proposed to use a latent-space EBM for controllable image synthesis. If a fixed generative model is used, then theoretically, any image-space EBM can be viewed as a latent-space EBM, as shown by [19] and [52]. We define the latent-space EBM following similar formulation as [49, 19, 52], with some new energy functions. We then extend the formulation

to incorporate the moment constraint [7], which was adopted for language modeling [37], but unlike [37], we define the moment constraint in the latent space to accommodate generative vision models.

We define a control $\mathcal{C}$ as $M$ independent properties $\{\boldsymbol{y}_1, \ldots, \boldsymbol{y}_M\}$, e.g., $\boldsymbol{y}_1$ can be a text description and $\boldsymbol{y}_2$ can be an attribute. The controllability can be defined as the EBM (detailed in Appendix B.1)

$$p(\boldsymbol{x}|\mathcal{C}) = \frac{p_{\boldsymbol{x}}(\boldsymbol{x})e^{-E_{\mathcal{C}}(\boldsymbol{x})}}{Z_X}, \quad E_{\mathcal{C}}(\boldsymbol{x}) = \sum_{i=1}^{M} \lambda_i E_i(\boldsymbol{x}, \boldsymbol{y}_i), Z_X = \int_{\boldsymbol{x}'} p_{\boldsymbol{x}}(\boldsymbol{x}')e^{-E_{\mathcal{C}}(\boldsymbol{x}')}d\boldsymbol{x}', \quad (1)$$

which reweights the image prior $p_{\boldsymbol{x}}(\boldsymbol{x})$ with energy $E_{\mathcal{C}}(\boldsymbol{x})$, where images with smaller energy are preferred. Using a pre-trained generative model $G : \mathbb{R}^d \to \mathcal{X}$ that maps a latent code $\boldsymbol{z}$ to an image $\boldsymbol{x}$, the image prior $p_{\boldsymbol{x}}(\boldsymbol{x})$ is defined by sampling a latent code $\boldsymbol{z}$ from $p_{\boldsymbol{z}} = \mathcal{N}(\boldsymbol{0}, \boldsymbol{I})$ and mapping it to $\boldsymbol{x} = G(\boldsymbol{z})$. Appendix B.2 shows that this EBM is equivalent to the latent-space EBM

$$p(\boldsymbol{z}|\mathcal{C}) = \frac{p_{\boldsymbol{z}}(\boldsymbol{z})e^{-E_{\mathcal{C}}(G(\boldsymbol{z}))}}{Z},$$

$$E_{\mathcal{C}}(G(\boldsymbol{z})) = \sum_{i=1}^{M} \lambda_i E_i(G(\boldsymbol{z}), \boldsymbol{y}_i), \quad Z = \int_{\boldsymbol{z}'} p_{\boldsymbol{z}}(\boldsymbol{z}')e^{-E_{\mathcal{C}}(G(\boldsymbol{z}'))}d\boldsymbol{z}'. \quad (2)$$

Latent-space EBM allows us to use any off-the-shelf model to specify the control. The following are some examples that are discussed in this paper (explained in Appendix B.1):

**Classifier energy:** Given a classifier $P(\cdot|\boldsymbol{x})$ and the target class $a$ that we want to sample images from, we define the classifier energy as $E_{\text{classifier}}(\boldsymbol{x}, a) = -\log P(a|\boldsymbol{x})$.

**CLIP energy:** Using the CLIP model [58], we define the CLIP energy as the cosine distance between the embeddings of the image and the text $\boldsymbol{t}$, averaged over $L$ differentiable augmentations [84, 46]:

$$E_{\text{CLIP}}(\boldsymbol{x}, \boldsymbol{t}) = \frac{1}{L} \sum_{l=1}^{L} \left( 1 - \cos \left\langle \text{CLIP}_{\text{img}}(\text{DiffAug}_l(\boldsymbol{x})), \text{CLIP}_{\text{text}}(\boldsymbol{t}) \right\rangle \right). \quad (3)$$

**Inverse graphics energy:** Given an inverse graphics model, $f_{\mathcal{X} \to \mathcal{P}}$, which infers image parameters (e.g., pose and expression), and the target parameters $\boldsymbol{\rho}$, we define the inverse graphics energy as

$$E_{\text{inv-graphics}}(\boldsymbol{x}, \boldsymbol{\rho}) = d\langle f_{\mathcal{X} \to \mathcal{P}}(\boldsymbol{x}), \boldsymbol{\rho} \rangle^2, \quad (4)$$

where $d\langle \cdot, \cdot \rangle$ is the geodesic distance between the inferred parameters and the target parameters.

**Moment constraint:** Some controls cannot be *directly* defined by off-the-shelf models, and the moment constraint [7, 37] is one of them. Given a mapping $\boldsymbol{\gamma} : \mathcal{X} \to \mathbb{R}^K$ (e.g., $\boldsymbol{\gamma}$ can be a classifier that outputs the probability simplex), the moment constraint defines the target distribution $p(\boldsymbol{x}|\mathcal{C})$ as

$$p(\boldsymbol{x}|\mathcal{C}) = \underbrace{\arg\min_{p(\boldsymbol{x}|\mathcal{C})} \mathbb{D}_{\text{KL}}\Big(p(\boldsymbol{x}|\mathcal{C}) \| p_{\boldsymbol{x}}(\boldsymbol{x})\Big)}_{\text{Deviation from the pre-trained distribution}}, \quad \text{s.t.} \quad \underbrace{\mathbb{E}_{\boldsymbol{x} \sim p(\boldsymbol{x}|\mathcal{C})}\big[\boldsymbol{\gamma}(\boldsymbol{x})\big] = \boldsymbol{\mu}}_{\text{Moment constraint}}, \quad (5)$$

where $\boldsymbol{\mu}$ is the user-specified vector. For example, if we want to generate images that are uniformly distributed across races, we may use a race classifier as $\boldsymbol{\gamma}$ and define $\boldsymbol{\mu} = \big(|\mathcal{A}|^{-1}, \ldots, |\mathcal{A}|^{-1}\big)$ where $\mathcal{A}$ is the set of races. It means that we would like to find a distribution that 1) stays close to the original, pre-trained generative model $G$ and 2) mitigates $G$'s bias. In this paper, we generalize the moment constraint to the latent space, and approximate the above objective as (detailed in Appendix B.6):

$$p(\boldsymbol{z}|\mathcal{C}) = \frac{p_{\boldsymbol{z}}(\boldsymbol{z}) \exp\left(\hat{\boldsymbol{\beta}}^{\top} \boldsymbol{\gamma}\big(G(\boldsymbol{z})\big)\right)}{Z}, \quad Z = \int_{\boldsymbol{z}'} p_{\boldsymbol{z}}(\boldsymbol{z}') \exp\left(\hat{\boldsymbol{\beta}}^{\top} \boldsymbol{\gamma}\big(G(\boldsymbol{z}')\big)\right)d\boldsymbol{z}', \quad (6)$$

$$\hat{\boldsymbol{\beta}} = \arg\min_{\boldsymbol{\beta}} \mathbb{E}_{\boldsymbol{z}^{(1)}, \ldots, \boldsymbol{z}^{(N)} \overset{\text{i.i.d.}}{\sim} p_{\boldsymbol{z}}(\boldsymbol{z}), \boldsymbol{x}^{(j)} = G(\boldsymbol{z}^{(j)})} \left\| \frac{\sum_{j=1}^{N} \exp\big(\boldsymbol{\beta}^{\top} \boldsymbol{\gamma}(\boldsymbol{x}^{(j)})\big) \boldsymbol{\gamma}(\boldsymbol{x}^{(j)})}{\sum_{j'=1}^{N} \exp\big(\boldsymbol{\beta}^{\top} \boldsymbol{\gamma}(\boldsymbol{x}^{(j')})\big)} - \boldsymbol{\mu} \right\|_2^2. \quad (7)$$

Notably, to optimize $\hat{\boldsymbol{\beta}}$ in Eq. (7), we leverage existing gradient-based optimization tools such as stochastic gradient descent (SGD) and Adam [38].

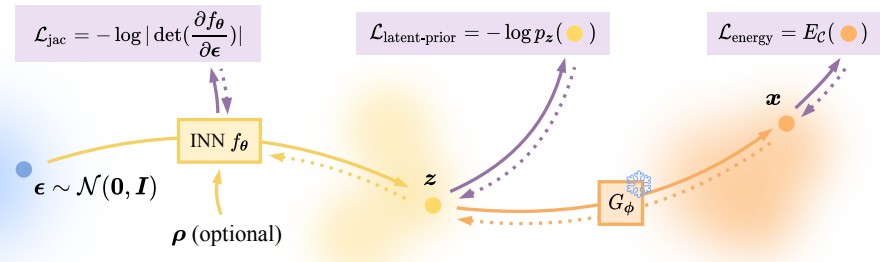

Figure 3: Illustration for Algorithm 2. In the forward pass (solid curves), we sample $\epsilon \sim \mathcal{N}(\mathbf{0}, \boldsymbol{I})$, map $\epsilon$ to latent code $\boldsymbol{z} = f_{\boldsymbol{\theta}}(\epsilon)$ with an INN $f_{\boldsymbol{\theta}}$, and map $\boldsymbol{z}$ to an image $\boldsymbol{x} = G(\boldsymbol{z})$ with a fixed generative model $G$. Dashed curves show the gradients. Details are provided in Section 3.2.

---

**Algorithm 2:** Approximating Latent-Space EBM with INN

**while** *not converged* **do**

> 1. Sample $\epsilon \sim \mathcal{N}(\mathbf{0}, \boldsymbol{I})$
> 2. Map $\epsilon$ to latent code $\boldsymbol{z} = f_{\boldsymbol{\theta}}(\epsilon)$
> 3. Map $\boldsymbol{z}$ to an image $\boldsymbol{x} = G(\boldsymbol{z})$
> 4. Optimize $\boldsymbol{\theta}$ with gradient $\nabla_{\boldsymbol{\theta}} \Big( -\log |\det(\frac{\partial f_{\boldsymbol{\theta}}}{\partial \epsilon})| - \log p_{\boldsymbol{z}}(\boldsymbol{z}) + E_{\mathcal{C}}(\boldsymbol{x}) \Big)$

---

## 3.2 Approximating EBM with Invertible Neural Network

Given the functional form of the EBM $p(\boldsymbol{z}|\mathcal{C})$, our next step is to approximate it with an efficient sampling network. To achieve this, we train a distribution $p_{\boldsymbol{\theta}}(\boldsymbol{z})$ that minimizes the KL divergence $\mathbb{D}_{\mathsf{KL}}(p_{\boldsymbol{\theta}}(\boldsymbol{z})\|p(\boldsymbol{z}|\mathcal{C}))$. Estimating $\mathbb{D}_{\mathsf{KL}}(p_{\boldsymbol{\theta}}(\boldsymbol{z})\|p(\boldsymbol{z}|\mathcal{C}))$ requires easily sampling $\boldsymbol{z} \sim p_{\boldsymbol{\theta}}$ and tractably computing $p_{\boldsymbol{\theta}}(\boldsymbol{z})$. Inspired by [54], we model $p_{\boldsymbol{\theta}}$ with an INN $f_{\boldsymbol{\theta}}$ that defines a bijection $\boldsymbol{z} = f_{\boldsymbol{\theta}}(\epsilon)$, which has two merits besides the invertibility: (1) one can easily sample $\boldsymbol{z}$ by sampling $\epsilon \sim \mathcal{N}(\mathbf{0}, \boldsymbol{I})$ and mapping it to $\boldsymbol{z} = f_{\boldsymbol{\theta}}(\epsilon)$, and (2) $p_{\boldsymbol{\theta}}(\boldsymbol{z})$ has a closed-form solution:

$$\log p_{\boldsymbol{\theta}}(\boldsymbol{z}) = \log \mathcal{N}(\epsilon|\mathbf{0}, \boldsymbol{I}) - \log |\det(\frac{\partial f_{\boldsymbol{\theta}}}{\partial \epsilon})|, \quad \boldsymbol{z} = f_{\boldsymbol{\theta}}(\epsilon). \tag{8}$$

Based on these properties of INN, we can rewrite our KL divergence objective $\mathbb{D}_{\mathsf{KL}}(p_{\boldsymbol{\theta}}(\boldsymbol{z})\|p(\boldsymbol{z}|\mathcal{C}))$ as (full derivations in Appendix B.3) the following form:

$$\mathbb{D}_{\mathsf{KL}}(p_{\boldsymbol{\theta}}(\boldsymbol{z})\|p(\boldsymbol{z}|\mathcal{C})) = \mathbb{E}_{\boldsymbol{z} \sim p_{\boldsymbol{\theta}}(\boldsymbol{z}), \boldsymbol{x}=G(\boldsymbol{z})} \Big[ \log \frac{p_{\boldsymbol{\theta}}(\boldsymbol{z})}{p_{\boldsymbol{z}}(\boldsymbol{z}) e^{-E_{\mathcal{C}}(\boldsymbol{x})}/Z} \Big]$$

$$= \mathbb{E}_{\epsilon \sim \mathcal{N}(\mathbf{0}, \boldsymbol{I}), \boldsymbol{z}=f_{\boldsymbol{\theta}}(\epsilon), \boldsymbol{x}=G(\boldsymbol{z})} \Big[ -\log |\det(\frac{\partial f_{\boldsymbol{\theta}}}{\partial \epsilon})| - \log p_{\boldsymbol{z}}(\boldsymbol{z}) + E_{\mathcal{C}}(\boldsymbol{x}) \Big] - \mathbb{H}_{\mathcal{N}(\mathbf{0}, \boldsymbol{I})} + \log Z. \tag{9}$$

Since $\mathbb{H}_{\mathcal{N}(\mathbf{0}, \boldsymbol{I})}$ and $\log Z$ are independent of $\boldsymbol{\theta}$, our training objective becomes

$$\arg \min_{\boldsymbol{\theta}} \mathbb{E}_{\epsilon \sim \mathcal{N}(\mathbf{0}, \boldsymbol{I}), \boldsymbol{z}=f_{\boldsymbol{\theta}}(\epsilon), \boldsymbol{x}=G(\boldsymbol{z})} \Big[ \underbrace{-\log |\det(\partial f_{\boldsymbol{\theta}}/\partial \epsilon)|}_{\mathcal{L}_{\text{jac}}} \underbrace{- \log p_{\boldsymbol{z}}(\boldsymbol{z})}_{\mathcal{L}_{\text{latent-prior}}} \underbrace{+ E_{\mathcal{C}}(\boldsymbol{x})}_{\mathcal{L}_{\text{energy}}} \Big]. \tag{10}$$

Figure 3 gives an illustration of our process, and Algorithm 2 describes the algorithmic details.

**PromptGen in a class-embedding space** (Figure 4(g)) Previous works [4, 64] have shown that *class conditioning* boosts generative models' performances on ImageNet [62]. Specifically, class-conditioned generative models map a latent code $\boldsymbol{z}$ and a class embedding $\boldsymbol{y}$ to $\boldsymbol{x} = G(\boldsymbol{z}, \boldsymbol{y})$. To extend PromptGen to these models, we train an INN $h_{\boldsymbol{\theta}}$ to map $\boldsymbol{\xi} \sim \mathcal{N}(\boldsymbol{\mu}, \boldsymbol{\sigma}^2 \boldsymbol{I})$ to $\boldsymbol{y} = h_{\boldsymbol{\theta}}(\boldsymbol{\xi})$, where $\boldsymbol{\mu}$ and $\boldsymbol{\sigma}$ are the mean and standard deviation of $G$'s class embeddings. The motivation for defining the distribution of $\boldsymbol{\xi}$ as $\mathcal{N}(\boldsymbol{\mu}, \boldsymbol{\sigma}^2 \boldsymbol{I})$ but not $\mathcal{N}(\mathbf{0}, \boldsymbol{I})$ is that we want the learned INN $h_{\boldsymbol{\theta}}$ to be volume-preserving, which is easier to train. The training objective is to minimize $\mathbb{D}_{\mathsf{KL}}(p_{\boldsymbol{\theta}}(\boldsymbol{z}, \boldsymbol{y})\|p(\boldsymbol{z}, \boldsymbol{y}|\mathcal{C}))$,

which is equivalent to (full derivations in Appendix B.4):

$$\arg\min_{\boldsymbol{\theta}} \mathbb{E}_{\boldsymbol{\epsilon}\sim\mathcal{N}(\boldsymbol{0},\boldsymbol{I}),\boldsymbol{\xi}\sim\mathcal{N}(\boldsymbol{\mu},\boldsymbol{\sigma}^2\boldsymbol{I}),\boldsymbol{z}=f_{\boldsymbol{\theta}}(\boldsymbol{\epsilon}),\boldsymbol{y}=h_{\boldsymbol{\theta}}(\boldsymbol{\xi}),\boldsymbol{x}=G(\boldsymbol{z},\boldsymbol{y})}\left[-\log|\det(\frac{\partial f_{\boldsymbol{\theta}}}{\partial\boldsymbol{\epsilon}})|-\log p_{\boldsymbol{z}}(\boldsymbol{z})\right.$$
$$\left.-\log|\det(\frac{\partial h_{\boldsymbol{\theta}}}{\partial\boldsymbol{\xi}})|-\log p_{\boldsymbol{y}}(\boldsymbol{y})+E_{\mathcal{C}}(\boldsymbol{x})\right], \tag{11}$$

where $p_{\boldsymbol{y}}(\boldsymbol{y}) = \mathcal{N}(\boldsymbol{y}|\boldsymbol{\mu},\boldsymbol{\sigma}^2\boldsymbol{I})$ is the estimated class-embedding distribution.

**PromptGen with conditional INN**  To generalize the control to continuous values, e.g., the scene parameters $\boldsymbol{\rho}$ in Eq. (4), we condition the INN on $\boldsymbol{\rho}$. The condition is modeled by replacing $\boldsymbol{z} = f_{\boldsymbol{\theta}}(\boldsymbol{\epsilon})$ with $\boldsymbol{z} = f_{\boldsymbol{\theta}}(\boldsymbol{\epsilon}, \boldsymbol{\rho})$. Space limited, we provide details and derivations in Appendix B.5. This extension results in a similar architecture to StyleFlow [1], but StyleFlow [1] uses MLE training on image-label pairs and is only applicable to explicit condition, which is not capable of modeling EBMs. When the condition is modeled by the equivariant operations proposed by [79], PromptGen also satisfies the homomorphism property [79] in the latent space.

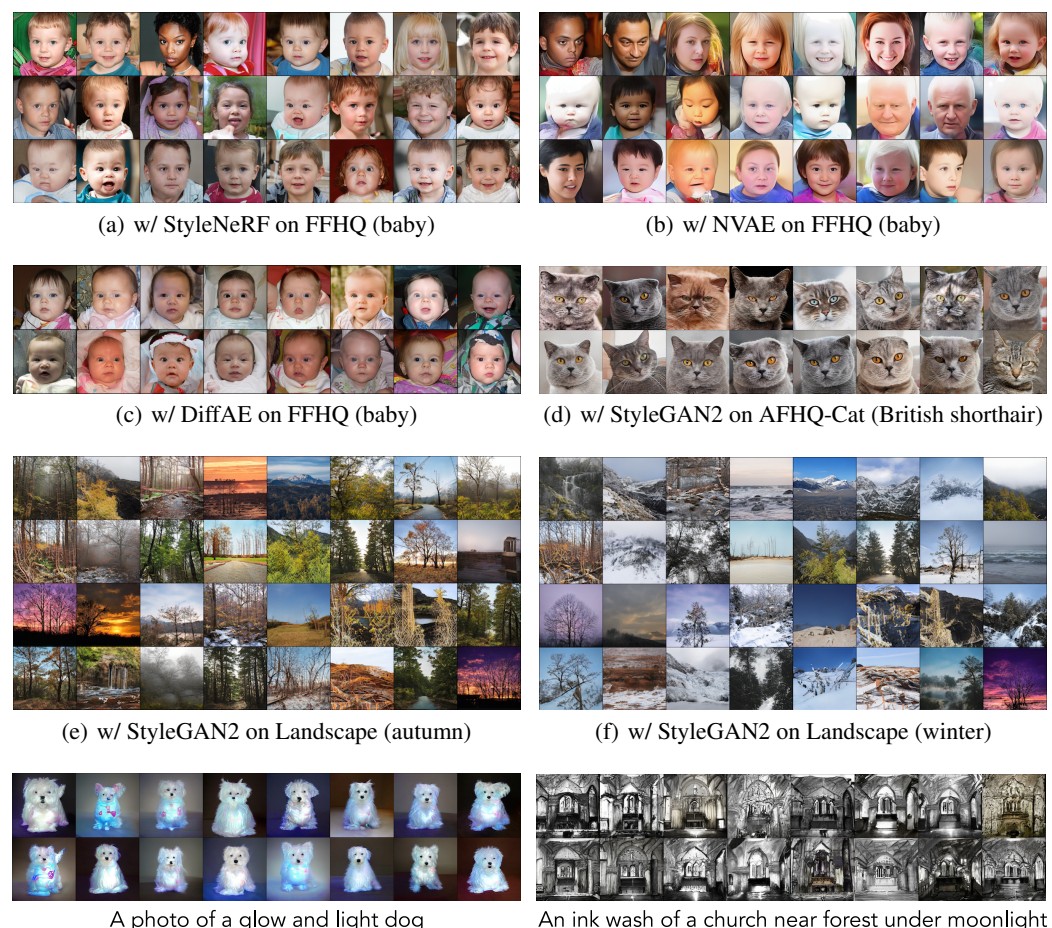

(a) w/ StyleNeRF on FFHQ (baby)

(b) w/ NVAE on FFHQ (baby)

(c) w/ DiffAE on FFHQ (baby)

(d) w/ StyleGAN2 on AFHQ-Cat (British shorthair)

(e) w/ StyleGAN2 on Landscape (autumn)

(f) w/ StyleGAN2 on Landscape (winter)

A photo of a glow and light dog

An ink wash of a church near forest under moonlight

(g) w/ BigGAN trained on ImageNet $512^2$

Figure 4: PromptGen is applicable to different generative models in various domains. Figure 4(a), Figure 4(b), and Figure 4(c) are PromptGen applied to StyleNeRF [21], NVAE [75], and diffusion autoencoder [57], with text description a photo of a baby. Figure 4(d) shows PromptGen applied to StyleGAN2 [36] trained on AFHQ-Cats [6], with text description a photo of a British shorthair cat Figure 4(e) and Figure 4(f) are PromptGen applied to StyleGAN2 trained on Landscape-HQ [70], with text descriptions a photo of {autumn, winter} scene. Figure 4(g) is the extension to the embedding space of BigGAN [4] on ImageNet [62]. All images are resized for visualization. See Appendix C for results on more datasets and text descriptions.

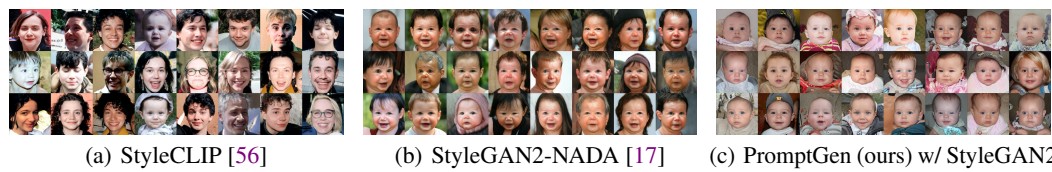

|  (a) StyleCLIP [56] | (b) StyleGAN2-NADA [17] | (c) PromptGen (ours) w/ StyleGAN2 |

Figure 5: Image synthesis based on text description, guided by the CLIP model. As the pre-trained generative model, we use StyleGAN2 trained on FFHQ $1024^2$ [35] with truncation $\psi = 0.7$. The text description used in this experiment is a photo of a baby. StyleCLIP requires optimization at inference, while StyleGAN2-NADA and PromptGen do not. All images are resized for visualization.

## 4    Experiments

This section describes the experimental validation of PromptGen. See Appendix A for experiments on synthetic data and Appendix C, D, E, and F for additional experiments on images and 3D meshes. Additional experimental details are provided in Appendix B.8.

### 4.1    Image Synthesis based on Text Description

This experiment illustrates the capability of PromptGen to sample images from a generative model driven by a text description $t$ using the pre-trained CLIP model [58] (the ViT-B/32 version). We used the CLIP energy from Eq. (3), with text descriptions such as a photo of a baby.

Figure 5 shows a comparison between PromptGen and two previous CLIP-guided image generation methods, StyleCLIP [56] and StyleGAN2-NADA [17]. We observe that PromptGen generates diverse and high-quality images of babies, while StyleCLIP struggles in controllability and image quality, and StyleGAN2-NADA generates baby-like adults.[2] These results show that (1) locally editing the latent code (i.e., StyleCLIP) is not always an effective method for controlling generative models (e.g., not all images' latent code can be locally edited into a baby); (2) domain adaptation (i.e., StyleGAN-NADA) is not effective in seeking modes in a generative model. Figure 4(a), Figure 4(b), and Figure 4(c) show how PromptGen applies to StyleNeRF [21], NVAE [75], and diffusion autoencoder (DiffAE; [57]). Although these generative models are all trained on FFHQ, baby images sampled from them have distinct characteristics. Figures 4(d)-4(f) show PromptGen applied to cats and landscape generation. Figure 4(g) shows the extension (Section 3.2) of our PromptGen to the class-embedding space of BigGAN [4], a class-conditional GAN trained on ImageNet [62]. We observe that PromptGen helps BigGAN generate images with complex text descriptions, which are out of BigGAN's training image distribution; however, the diversity seems to be limited in this extension.

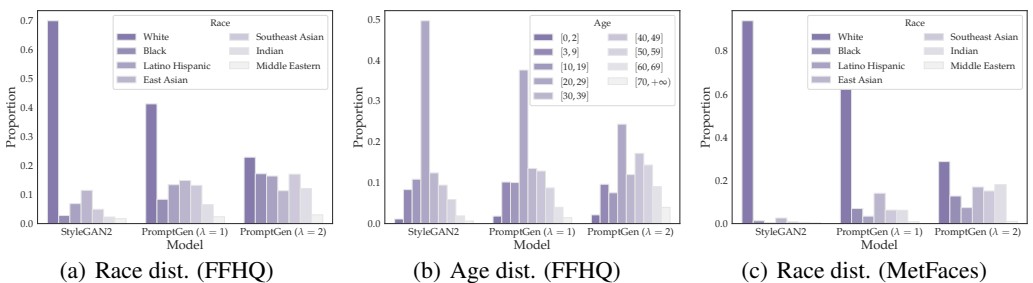

|  (a) Race dist. (FFHQ) | (b) Age dist. (FFHQ) | (c) Race dist. (MetFaces) |

Figure 6: With the moment constraint, PromptGen de-biases StyleGAN2 (on FFHQ and MetFaces $1024^2$, truncation $\psi = 0.7$). See Appendix D for image samples and Table 5 for quantitative results.

### 4.2    De-Biasing Pre-Trained Generative Models

An important problem in generative models is to generate fair distributions w.r.t a set of attributes of interest. For instance, Figure 6(a) shows that StyleGAN2 generates images with bias across races and

---

[2]Since we care about distributions, ***none*** *of the images in this paper are cherry- or lemon-picked.*

ages. PromptGen de-biases StyleGAN2 models trained on FFHQ $1024^2$ and MetFaces $1024^2$ [33] in terms of *categorical* attributes, using the moment constraint defined in Eq. (5) and Eq. (6). As control, we used a classifier trained on FairFace $224^2$ [32] as $\gamma$. We defined $\boldsymbol{\mu} = \left(|\mathcal{A}|^{-1}, \ldots, |\mathcal{A}|^{-1}\right)$, where $\mathcal{A}$ is the set of races. Similar to the energy weights $\lambda_i$ defined in Eq. (2), we propose to rescale the trained $\hat{\boldsymbol{\beta}}$ as $\lambda\hat{\boldsymbol{\beta}}$. Figure 6 shows that PromptGen de-biases the race and age effectively.

Existing de-biasing baselines consider *binary* attributes [31]. For a fair comparison with them, we adopted their setting to use binary classifiers trained on CelebA [47] for de-biasing and evaluation. Since classifiers trained on CelebA also suffer from the spurious correlation between attributes, we did *not* use the moment constraint for this experiment. Instead, since PromptGen allows conditional image generation with the classifier energy, we generated the same number of samples conditioned on each attribute or attribute combination. Table 2 shows that PromptGen has competitive performance on de-biasing for attributes and attribute combinations.

Table 2: Comparison with baselines for de-biasing binary attributes and their correlations. Following [31], we use classifiers on CelebA [47]. Baseline performances are copied from [31]. PromptGen has competitive performance, even in the cases where FairStyle achieves nearly perfect performance.

| | FFHQ (binary attributes) | | | | | |
|---|---|---|---|---|---|---|
| | $\mathbb{D}_{\text{KL}}^{\text{gender}}\downarrow$ | $\mathbb{D}_{\text{KL}}^{\text{eyeglasses}}\downarrow$ | $\mathbb{D}_{\text{KL}}^{\text{blond hair}}\downarrow$ | $\mathbb{D}_{\text{KL}}^{\text{age+gender}}\downarrow$ | $\mathbb{D}_{\text{KL}}^{\text{age+eyeglasses}}\downarrow$ | $\mathbb{D}_{\text{KL}}^{\text{gender+eyeglasses}}\downarrow$ |
| FFHQ (real data) | 0.015 | 0.186 | – | 0.246 | 0.355 | 0.242 |
| StyleGAN2 [36] | 0.018 | 0.180 | – | 0.279 | 0.384 | 0.250 |
| StyleFlow [1] | 0.023 | 0.061 | – | 0.214 | 0.162 | 0.121 |
| FairGen [72] | $4.21 \times 10^{-4}$ | $7.07 \times 10^{-4}$ | – | 0.0373 | 0.0330 | 0.00185 |
| FairStyle [31] | $\mathbf{3.20} \times 10^{-7}$ | $\mathbf{0}$ | – | 0.0257 | 0.0157 | $\mathbf{0.000241}$ |
| PromptGen (ours) | $1.71 \times 10^{-5}$ | $1.72 \times 10^{-5}$ | 0.0008 | $\mathbf{0.000558}$ | $\mathbf{0.000415}$ | 0.000628 |

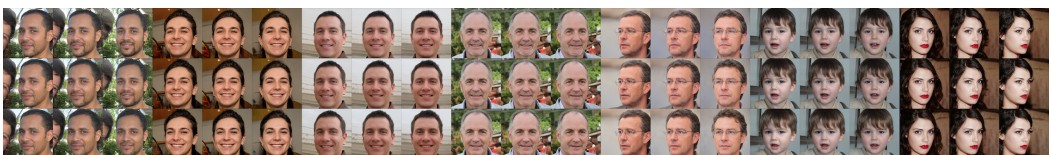

Figure 7: Using inverse graphics model DECA [16] and scene parameters $\boldsymbol{\rho}_{\text{pose}} \in \text{SO}(3)$, PromptGen controls the pose of StyleGAN2 while preserving the identity. All images are resized for visualization.

Table 3: Pose-controlled face generation. In this experiment, PromptGen uses StyleGAN2 as the pre-trained generative model. Results for GRAF [66], pi-GAN [5], GIRAFFE [53], and StyleNeRF [21] are from [21]; results for DiscoFaceGAN (DFG) [9] and GAN-Control [69] are from [69].

| | StyleGAN2 | GRAF | pi-GAN | GIRAFFE | StyleNeRF | DFG | GAN-Control | PromptGen |
|---|---|---|---|---|---|---|---|---|
| Resolution | $1024^2$ | $256^2$ | $256^2$ | $256^2$ | $1024^2$ | $256^2$ | $512^2$ | $1024^2$ |
| Pose | ✗ | ✓ | ✓ | ✓ | ✓ | ✓ | ✓ | ✓ |
| FID↓ | 3 | 71 | 85 | 35 | 8 | 13 | 6 | 4 |
| Dist. w/ same ID↓ | – | – | – | – | – | 0.83 | 0.68 | 0.45 |
| Dist. w/ diff. ID | – | – | – | – | – | 1.73 | 1.90 | 1.37 |

### 4.3 Pose-Guided Face Synthesis

With an inverse graphics model, PromptGen can control the pose of faces generated by StyleGAN2. We used the DECA model [16], which infers the parameters of FLAME [44], a parametric facial graphics model. We set $\boldsymbol{\rho} \in \text{SO}(3)$ as FLAME's three neck poses and used the conditional INN extension introduced in Section 3.2. To enable generating different poses of the same identity (ID), we propose ID energy using the IR-SE50 model [8]. Specifically, given a canonical pose $\boldsymbol{\rho}_0$, we define $\boldsymbol{z}_0 = f_{\boldsymbol{\theta}}(\boldsymbol{\epsilon}, \boldsymbol{\rho}_0)$, and the ID energy is defined as (detailed in Appendix B.7)

$$E_{\text{ID}}(\boldsymbol{x}_0, \boldsymbol{x}) = 1 - \cos\left\langle R(\boldsymbol{x}_0), R(\boldsymbol{x})\right\rangle, \quad \boldsymbol{x}_0 = G(\boldsymbol{z}_0), \boldsymbol{x} = G(\boldsymbol{z}), \tag{12}$$

where $R$ is the IR-SE50 model [8] that computes face embeddings. Figure 7 shows that PromptGen generates faces of the same ID in different poses, even without being explicitly trained with poses as conditions. We computed the FID score [24] of each model using Clean-FID [55]; following [36], we did not use the truncation trick when computing the FID score. Table 3 shows that PromptGen outperforms existing models in terms of the FID score. Following [69], we then reported the average IR-SE50 [8] embedding distances for images with the same ID and with different IDs. Results show that PromptGen achieves the best ID preservation, with a slight sacrifice of ID diversity.

## 4.4 Iterative Distributional Control via Functional Composition

This section discusses an interesting bias that PromptGen reveals about the CLIP model. Figure 8(a) shows images generated by PromptGen (with StyleGAN2) with the CLIP model and the text description a photo of a person without makeup, where more females are generated than males. This bias should not be attributed to *image pre-training data* since images contain a bias in the opposite direction, i.e., men are less likely to have makeup. We argue that this bias should be explained by CLIP having learned a "reporting bias" in *vision-language pre-training data*: people are more likely to *say* "a person without makeup" when the person is a female (detailed analysis in Appendix F).

Besides revealing the above "reporting bias", PromptGen can also mitigate this bias via an iterative control, enabled by the functional composition in Algorithm 1. Specifically, in the second iteration, we de-biased the gender distribution of $G \circ f_{\boldsymbol{\theta}}$ instead of $G$, where $f_{\boldsymbol{\theta}}$ is the INN learned for the text control. For de-biasing, we used the moment constraint with $\hat{\boldsymbol{\beta}}$ trained for $G \circ f_{\boldsymbol{\theta}}$. Figure 8(b) shows that females and males are uniformly distributed after the moment constraint in the second iteration.

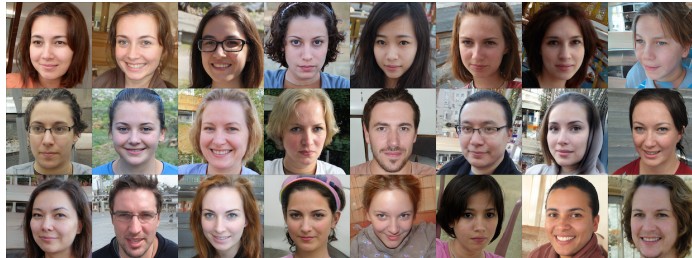

(a) PromptGen (iteration 1) with text a photo of a person without makeup. Gender distribution: female: 81.6%; male: 18.4%.

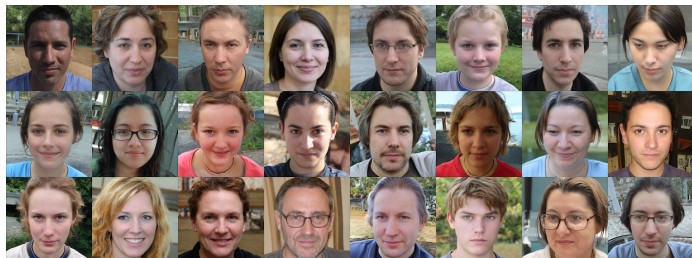

(b) PromptGen (iteration 2). Gender de-biasing ($\lambda = 2$) for the distribution learned in iteration 1. Gender distribution: female: 49.3%; male: 50.7%.

Figure 8: A curious case of the CLIP model: with description a photo of a person without makeup, PromptGen generates more female images than male, showing that CLIP learns a "reporting bias". Besides revealing this "reporting bias", PromptGen can also mitigate this bias via an iterative control, allowing us to de-bias the text-controlled distribution. All images are resized for visualization.

## 4.5 Inference Latency

We compared our PromptGen and the plug-and-play generative model (PPGM) [49] in terms of the inference latency. PPGM uses Langevin dynamics [76] to optimize over the latent space at inference, while PromptGen samples images in a feed-forward manner. In this experiment, we used PPGM and our PromptGen to approximate the same EBM with CLIP energy. Inference times were estimated on

an NVIDIA RTX A4000 GPU. Table 4 shows that our PromptGen has the highest performance and efficiency. Compared to our PromptGen, PPGM requires $100\times$ inference time to achieve comparable results. Moreover, learning $f_\theta$ in PromptGen requires training-time optimization, and this amortized optimization is useful when one wants to reuse a controlled distribution many times.

Table 4: Comparison between PromptGen and PPGM, when approximating the *same* EBM with CLIP energy. $n$ is the number of inference-time optimization steps used by PPGM. All models used Style-GAN2 as the pre-trained generative model. The text descriptions are a photo of a {baby, boy, girl}.

|  | PPGM ($n = 10$) | PPGM ($n = 50$) | PromptGen (ours) |
|---|---|---|---|
| CLIP energy (baby)↓ | 0.7327 | 0.7134 | **0.7038** |
| CLIP energy (girl)↓ | 0.7257 | **0.7184** | 0.7199 |
| CLIP energy (boy)↓ | 0.7263 | 0.7114 | **0.7081** |
| Inference time per sample (sec.)↓ | 4.4 | 21.5 | **0.2** |
| Back-propagation through CLIP at inference | 10 | 50 | 0 |
| When is it equal to the EBM? | $n \to \infty$ | $n \to \infty$ | $\mathbb{D}_{\mathsf{KL}}(p_\theta(z)\|p(z|\mathcal{C})) = 0$ |

## 5   Conclusions and Future Work

This paper proposes PromptGen, a unified framework to learn latent distributions for distributional control of pre-trained generative models, by leveraging the knowledge of other off-the-shelf models. Unlike previous approaches, which require optimization for generating each sample at inference, PromptGen trains an invertible neural network to approximate the latent distribution. Thus, Prompt-Gen can be viewed as an amortization of the inference-time optimization into a feed-forward neural network, allowing for efficient, feed-forward sampling at inference. Further, this amortization also makes PromptGen not require the availability of those off-the-shelf models at inference. In practice, PromptGen offers generality for algorithmic design and modularity for control composition, and it also enables iterative controls via functional composition. Experiments validate that PromptGen applies to various generative models (StyleGAN2, StyleNeRF, diffusion autoencoder, and NVAE), control types (continuous, discrete, and moment constraint), off-the-shelf models (CLIP, classifiers, and inverse graphics models), and data domains (faces, churches, animals, ImageNet, and landscapes).

**Limitation and future work:**   We provide an error analysis in Appendix F and a discussion on societal impact in Section 6. PromptGen is restricted by the pre-trained generative model's coverage [3, 28]. Notably, since PromptGen focuses on mode-seeking and reweighting instead of domain adaptation, the support set of the output distribution will not change after PromptGen learning. We find it interesting to further explore combining PromptGen and domain adaptation [17] to achieve better generalization and adaptability. PromptGen depends on the off-the-shelf models that provide knowledge about the control (Appendix F); it can be beneficial to explore how to learn energy functions [15, 27] (besides our moment constraint) for fine-grained control with less bias.

## 6   Societal Impact

Like any technology, PromptGen has both benefits and drawbacks for society. On the positive side, we have demonstrated that PromptGen can uncover biases learned by text-image models (like CLIP) and to de-bias text-image models and pre-trained generative models, suggesting that PromptGen may be a helpful tool for fair AI if used properly. The efficient inference of PromptGen also helps reduce the computational expense, giving a positive impact on the environment. However, improved controllability makes it easier to synthesize targeted images; for instance, the creation of deceptive media such as DeepFakes [77, 74] and privacy leakage. To battle these cases, we expect researchers and practitioners to use technologies that can detect fake media and mitigate privacy leakage. We also encourage practitioners to consider these risks when using PromptGen to develop systems.

## Acknowledgments and Disclosure of Funding

The authors would like to thank the anonymous reviewers, Zoltán Ádám Milacski, Jianchun Chen, and Shubhra Aich for their valuable feedback on drafts of this paper.

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
