

Figure 9: PromptGen on synthetic data. (a) The real distribution is a Gaussian mixture distribution biased towards one mode. We derive a closed-form fair classifier. (b) A GAN trained on this real distribution. (c) With the fair classifier as control, PromptGen learns a distribution concentrated in a specific region. (d) We derive the moment constraint following Section 3.1 and Section 4.2, and PromptGen learns to up-weight the under-represented regions.

## A  PromptGen on Synthetic Data

We demonstrate the behavior of our PromptGen with two-dimensional synthetic data, using GAN as the generative model. This synthetic experiment is illustrated in Figure 9. Specifically, we created a "real" distribution that is a Gaussian mixture distribution biased towards one mode. We derive a closed-form fair classifier based on this distribution, detailed in the purple block in Figure 9. When a GAN is trained to approximate this distribution, its outputs are biased towards the over-represented mode, as shown by Figure 9(b). Figure 9(c) illustrates the controllability experiment: using the fair classifier as control, PromptGen learns a distribution concentrated in a specific region of the output space. Figure 9(d) illustrates the de-biasing experiment: using the moment constraint (Section 3.1 and Section 4.2), PromptGen upweights the under-represented regions of the output space.

## B  Method Details and Derivations

### B.1  Controllability as EBM

In Section 3, we defined a control $\mathcal{C}$ as $M$ independent properties $\{\boldsymbol{y}_1, \ldots, \boldsymbol{y}_M\}$. For example, $\boldsymbol{y}_1$ can be a text description, and $\boldsymbol{y}_2$ can be an attribute. In this part, we first elaborate on why controllability can be formed as EBMs in Eq. (1). We then provide concrete examples of the distributions derived from different energy functions defined in Section 3.1.

We denote the image prior as $p_{\boldsymbol{x}}(\boldsymbol{x})$, the only distribution that can be estimated from data when labels for the control are not provided during generative model pre-training. Given the control $\mathcal{C} = \{\boldsymbol{y}_1, \ldots, \boldsymbol{y}_M\}$, we resort to Bayes' theorem to rewrite the conditional distribution $p(\boldsymbol{x}|\mathcal{C})$ as

$$p(\boldsymbol{x}|\mathcal{C}) \propto p_{\boldsymbol{x}}(\boldsymbol{x})p(\mathcal{C}|\boldsymbol{x}) \tag{13}$$

$$= p_{\boldsymbol{x}}(\boldsymbol{x})p(\boldsymbol{y}_1|\boldsymbol{x})\prod_{i=2}^{M} p(\boldsymbol{y}_i|\boldsymbol{x}, \boldsymbol{y}_{<i}) \tag{14}$$

$$= p_{\boldsymbol{x}}(\boldsymbol{x})\prod_{i=1}^{M} p(\boldsymbol{y}_i|\boldsymbol{x}) \quad \text{(independence assumption).} \tag{15}$$

For Eq. (15) to be well-defined, we need to define $p(\boldsymbol{y}_i|\boldsymbol{x})$ for each $\boldsymbol{y}_i$. In order to incorporate the knowledge of arbitrary off-the-self models (besides image classifiers), we define each $p(\boldsymbol{y}_i|\boldsymbol{x})$ as

$$p(\boldsymbol{y}_i|\boldsymbol{x}) = \frac{\exp(-\lambda_i E_i(\boldsymbol{x}, \boldsymbol{y}_i))}{Z_i}, \quad Z_i = \int_{\boldsymbol{y}_i'} \exp(-\lambda_i E_i(\boldsymbol{x}, \boldsymbol{y}_i'))d\boldsymbol{y}_i'. \tag{16}$$

Eq. (16) says that $p(\boldsymbol{y}_i|\boldsymbol{x})$ is proportional to the exponential of an energy function $E_i(\boldsymbol{x}, \boldsymbol{y}_i)$, where $\boldsymbol{y}_i$ with lower energy has higher density or mass. Note that Eq. (16) defines a distribution over all possible values of $\boldsymbol{y}_i$ instead of all possible values of $\boldsymbol{x}$. Combining Eq. (15) and Eq. (16), we have

$$p(\boldsymbol{x}|\mathcal{C}) = \frac{p_{\boldsymbol{x}}(\boldsymbol{x})e^{-E_{\mathcal{C}}(\boldsymbol{x})}}{Z_X}, \quad E_{\mathcal{C}}(\boldsymbol{x}) = \sum_{i=1}^{M} \lambda_i E_i(\boldsymbol{x}, \boldsymbol{y}_i), Z_X = \int_{\boldsymbol{x}'} p_{\boldsymbol{x}}(\boldsymbol{x}')e^{-E_{\mathcal{C}}(\boldsymbol{x}')}d\boldsymbol{x}', \quad (17)$$

which is the same equation as Eq. (1).

In the following, we use Eq. (16) to derive the distributions from the classifier energy, CLIP energy, and inverse graphics energy defined in Section 3.1.

**Classifier energy:** Given a classifier $P(\cdot|\boldsymbol{x})$ and the target class $a$, we define the classifier energy as $E_{\text{classifier}}(\boldsymbol{x}, a) = -\log P(a|\boldsymbol{x})$. Using Eq. (16), we arrive at:

$$p_{\text{classifier}}(a|\boldsymbol{x}) = \frac{\exp(\lambda_{\text{classifier}} \log P(a|\boldsymbol{x}))}{\sum_{a'} \exp(\lambda_{\text{classifier}} \log P(a'|\boldsymbol{x}))} = \frac{P(a|\boldsymbol{x})^{\lambda_{\text{classifier}}}}{\sum_{a'} P(a'|\boldsymbol{x})^{\lambda_{\text{classifier}}}}, \quad (18)$$

which is equivalent to a temperature-adjusted distribution of the original classifier.

**CLIP energy:** Using Eq. (16), the CLIP energy in Eq. (3) is equivalent to

$$p_{\text{CLIP}}(\boldsymbol{t}|\boldsymbol{x}) \propto \exp\left( -\frac{\lambda_{\text{CLIP}}}{L} \sum_{l=1}^{L} \left( 1 - \cos\left\langle \text{CLIP}_{\text{img}}\big(\text{DiffAug}_l(\boldsymbol{x})\big), \text{CLIP}_{\text{text}}(\boldsymbol{t})\right\rangle \right) \right). \quad (19)$$

**Inverse graphics energy:** Using Eq. (16), the inverse graphics energy in Eq. (4) is equivalent to

$$p_{\text{inv-graphics}}(\boldsymbol{\rho}|\boldsymbol{x}) \propto \exp(-\lambda_{\text{inv-graphics}}d\langle f_{\mathcal{X}\to\mathcal{P}}(\boldsymbol{x}), \boldsymbol{\rho}\rangle^2). \quad (20)$$

If the geodesic distance $d\langle\cdot, \cdot\rangle$ is the Euclidean distance, then $p_{\text{inv-graphics}}(\boldsymbol{\rho}|\boldsymbol{x})$ is a Gaussian distribution whose mean is $\boldsymbol{\rho}$ and whose variance depends on the hyperparameter $\lambda_{\text{inv-graphics}}$; if the geodesic distance $d\langle\cdot, \cdot\rangle$ is the spherical distance (e.g., the distance between pose parameters defined on a unit sphere), then $p_{\text{inv-graphics}}(\boldsymbol{\rho}|\boldsymbol{x})$ is a vMF distribution.

## B.2   Equivalence between Image-Space EBM and Latent-Space EBM

**Proposition 1.** *Define $\boldsymbol{x} \sim p_{\boldsymbol{x}}(\boldsymbol{x})$ as $\underline{\boldsymbol{z} \sim p_{\boldsymbol{z}}(\boldsymbol{z}), \boldsymbol{x} = G(\boldsymbol{z})}$ and $p(\boldsymbol{x}|\mathcal{C})$ as $\underline{\boldsymbol{z} \sim p(\boldsymbol{z}|\mathcal{C}), \boldsymbol{x} = G(\boldsymbol{z})}$, where $p(\boldsymbol{z}|\mathcal{C})$ is defined as the following EBM:*

$$p(\boldsymbol{z}|\mathcal{C}) = \frac{p_{\boldsymbol{z}}(\boldsymbol{z})e^{-E_{\mathcal{C}}(G(\boldsymbol{z}))}}{Z}, \quad E_{\mathcal{C}}(\boldsymbol{x}) = \sum_{i=1}^{M} \lambda_i E_i(\boldsymbol{x}, \boldsymbol{y}_i), Z = \int_{\boldsymbol{z}'} p_{\boldsymbol{z}}(\boldsymbol{z}')e^{-E_{\mathcal{C}}(G(\boldsymbol{z}'))}d\boldsymbol{z}'. \quad (21)$$

*We have*

$$p(\boldsymbol{x}|\mathcal{C}) = \frac{p_{\boldsymbol{x}}(\boldsymbol{x})e^{-E_{\mathcal{C}}(\boldsymbol{x})}}{Z_X}, \quad Z_X = \int_{\boldsymbol{x}'} p_{\boldsymbol{x}}(\boldsymbol{x}')e^{-E_{\mathcal{C}}(\boldsymbol{x}')}d\boldsymbol{x}'. \quad (22)$$

In spirit, our proof follows the proof in [52], which follows [19]. The difference between our proof and that in [52] is that we derive $p(\boldsymbol{x}|\mathcal{C})$ from $p(\boldsymbol{z}|\mathcal{C})$ while they derived $p(\boldsymbol{z}|\mathcal{C})$ from $p(\boldsymbol{x}|\mathcal{C})$.

*Proof.* Based on Lemma 1 in [19] and Lemma 1 in [52], $p(\boldsymbol{z}|\mathcal{C})$ is equivalent to rejection sampling with proposal distribution $p_{\boldsymbol{z}}(\boldsymbol{z})$ and acceptance probability

$$r(\boldsymbol{z}) = \frac{e^{-E_{\mathcal{C}}(G(\boldsymbol{z}))}}{M_{\mathcal{C}} \cdot Z}, \quad \text{where } \forall \boldsymbol{z}, M_{\mathcal{C}} > \frac{e^{-E_{\mathcal{C}}(G(\boldsymbol{z}))}}{Z}. \quad (23)$$

Since $p(\boldsymbol{x}|\mathcal{C})$ is defined as $\underline{\boldsymbol{z} \sim p_{\boldsymbol{z}}(\boldsymbol{z}|\mathcal{C}), \boldsymbol{x} = G(\boldsymbol{z})}$, $p(\boldsymbol{x}|\mathcal{C})$ is equivalent to rejection sampling with proposal distribution $\underline{p_{\boldsymbol{z}}(\boldsymbol{z}), \boldsymbol{x} = G(\boldsymbol{z})}$ and acceptance probability

$$r(\boldsymbol{x}) = \frac{e^{-E_{\mathcal{C}}(\boldsymbol{x})}}{M_{\mathcal{C}} \cdot Z}. \quad (24)$$

Note that the above proposal distribution $\boldsymbol{z} \sim p_{\boldsymbol{z}}(\boldsymbol{z}), \boldsymbol{x} = G(\boldsymbol{z})$ is the same as $p_{\boldsymbol{x}}(\boldsymbol{x})$ by definition. Based on Lemma 1 in [19] and Lemma 1 in [52], we arrive at:

$$p(\boldsymbol{x}|\mathcal{C}) = \frac{p_{\boldsymbol{x}}(\boldsymbol{x})r(\boldsymbol{x})}{\mathbb{E}_{\boldsymbol{x}' \sim p_{\boldsymbol{x}}(\boldsymbol{x}')}[r(\boldsymbol{x}')]} \tag{25}$$

$$= \frac{p_{\boldsymbol{x}}(\boldsymbol{x})e^{-E_{\mathcal{C}}(\boldsymbol{x})}/(M_{\mathcal{C}} \cdot Z)}{\mathbb{E}_{\boldsymbol{x}' \sim p_{\boldsymbol{x}}(\boldsymbol{x}')}[e^{-E_{\mathcal{C}}(\boldsymbol{x}')}/(M_{\mathcal{C}} \cdot Z)]} \tag{26}$$

$$= \frac{p_{\boldsymbol{x}}(\boldsymbol{x})e^{-E_{\mathcal{C}}(\boldsymbol{x})}}{\mathbb{E}_{\boldsymbol{x}' \sim p_{\boldsymbol{x}}(\boldsymbol{x}')}[e^{-E_{\mathcal{C}}(\boldsymbol{x}')}]} \tag{27}$$

$$= \frac{p_{\boldsymbol{x}}(\boldsymbol{x})e^{-E_{\mathcal{C}}(\boldsymbol{x})}}{Z_X}, \quad Z_X = \int_{\boldsymbol{x}'} p_{\boldsymbol{x}}(\boldsymbol{x}')e^{-E_{\mathcal{C}}(\boldsymbol{x}')}d\boldsymbol{x}'. \tag{28}$$

$\square$

## B.3  Derivation of Eq. (9): Approximating EBM with INN

The full derivation of Eq. (9) is given by:

$$\mathbb{D}_{\mathsf{KL}}(p_{\boldsymbol{\theta}}(\boldsymbol{z})\|p(\boldsymbol{z}|\mathcal{C}))$$
$$= \mathbb{E}_{\boldsymbol{z} \sim p_{\boldsymbol{\theta}}(\boldsymbol{z})}\Big[\log \frac{p_{\boldsymbol{\theta}}(\boldsymbol{z})}{p(\boldsymbol{z}|\mathcal{C})}\Big]$$
$$= \mathbb{E}_{\boldsymbol{z} \sim p_{\boldsymbol{\theta}}(\boldsymbol{z}),\boldsymbol{x}=G(\boldsymbol{z})}\Big[\log \frac{p_{\boldsymbol{\theta}}(\boldsymbol{z})}{p_{\boldsymbol{z}}(\boldsymbol{z})e^{-E_{\mathcal{C}}(\boldsymbol{x})}/Z}\Big]$$
$$= \mathbb{E}_{\boldsymbol{\epsilon} \sim \mathcal{N}(\mathbf{0},\boldsymbol{I}),\boldsymbol{z}=f_{\boldsymbol{\theta}}(\boldsymbol{\epsilon}),\boldsymbol{x}=G(\boldsymbol{z})}\Big[\log \mathcal{N}(\boldsymbol{\epsilon}|\mathbf{0},\boldsymbol{I}) - \log |\det(\frac{\partial f_{\boldsymbol{\theta}}}{\partial \boldsymbol{\epsilon}})| \tag{29}$$
$$- \log p_{\boldsymbol{z}}(\boldsymbol{z}) + E_{\mathcal{C}}(\boldsymbol{x}) + \log Z\Big]$$
$$= \mathbb{E}_{\boldsymbol{\epsilon} \sim \mathcal{N}(\mathbf{0},\boldsymbol{I}),\boldsymbol{z}=f_{\boldsymbol{\theta}}(\boldsymbol{\epsilon}),\boldsymbol{x}=G(\boldsymbol{z})}\Big[-\log |\det(\frac{\partial f_{\boldsymbol{\theta}}}{\partial \boldsymbol{\epsilon}})| - \log p_{\boldsymbol{z}}(\boldsymbol{z})$$
$$+ E_{\mathcal{C}}(\boldsymbol{x})\Big] - \mathbb{H}_{\mathcal{N}(\mathbf{0},\boldsymbol{I})} + \log Z.$$

## B.4  Derivation of Eq. (11): Extension to Generative Models with a Class-Embedding Space

The full derivation of Eq. (11) is given by:

$$\mathbb{D}_{\mathsf{KL}}(p_{\boldsymbol{\theta}}(\boldsymbol{z},\boldsymbol{y})\|p(\boldsymbol{z},\boldsymbol{y}|\mathcal{C}))$$
$$= \mathbb{E}_{(\boldsymbol{z},\boldsymbol{y}) \sim p_{\boldsymbol{\theta}}(\boldsymbol{z},\boldsymbol{y})}\Big[\log \frac{p_{\boldsymbol{\theta}}(\boldsymbol{z},\boldsymbol{y})}{p(\boldsymbol{z},\boldsymbol{y}|\mathcal{C})}\Big]$$
$$= \mathbb{E}_{(\boldsymbol{z},\boldsymbol{y}) \sim p_{\boldsymbol{\theta}}(\boldsymbol{z},\boldsymbol{y}),\boldsymbol{x}=G(\boldsymbol{z},\boldsymbol{y})}\Big[\log \frac{p_{\boldsymbol{\theta}}(\boldsymbol{z},\boldsymbol{y})}{p_{\boldsymbol{z},\boldsymbol{y}}(\boldsymbol{z},\boldsymbol{y})e^{-E_{\mathcal{C}}(\boldsymbol{x})}/Z}\Big]$$
$$= \mathbb{E}_{(\boldsymbol{z},\boldsymbol{y}) \sim p_{\boldsymbol{\theta}}(\boldsymbol{z},\boldsymbol{y}),\boldsymbol{x}=G(\boldsymbol{z},\boldsymbol{y})}\Big[\log \frac{p_{\boldsymbol{\theta}}(\boldsymbol{z},\boldsymbol{y})}{p_{\boldsymbol{z}}(\boldsymbol{z})p_{\boldsymbol{y}}(\boldsymbol{y})e^{-E_{\mathcal{C}}(\boldsymbol{x})}/Z}\Big] \quad (\boldsymbol{z} \text{ and } \boldsymbol{y} \text{ are independent})$$
$$= \mathbb{E}_{\boldsymbol{\epsilon} \sim \mathcal{N}(\mathbf{0},\boldsymbol{I}),\boldsymbol{\xi} \sim \mathcal{N}(\boldsymbol{\mu},\boldsymbol{\sigma}^2\boldsymbol{I}),\boldsymbol{z}=f_{\boldsymbol{\theta}}(\boldsymbol{\epsilon}),\boldsymbol{y}=h_{\boldsymbol{\theta}}(\boldsymbol{\xi}),\boldsymbol{x}=G(\boldsymbol{z},\boldsymbol{y})}\Big[\log \mathcal{N}(\boldsymbol{\epsilon}|\mathbf{0},\boldsymbol{I}) - \log |\det(\frac{\partial f_{\boldsymbol{\theta}}}{\partial \boldsymbol{\epsilon}})| \tag{30}$$
$$+ \log \mathcal{N}(\boldsymbol{\xi}|\boldsymbol{\mu},\boldsymbol{\sigma}^2\boldsymbol{I}) - \log |\det(\frac{\partial h_{\boldsymbol{\theta}}}{\partial \boldsymbol{\xi}})| - \log p_{\boldsymbol{z}}(\boldsymbol{z}) - \log p_{\boldsymbol{y}}(\boldsymbol{y}) + E_{\mathcal{C}}(\boldsymbol{x}) + \log Z\Big]$$
$$= \mathbb{E}_{\boldsymbol{\epsilon} \sim \mathcal{N}(\mathbf{0},\boldsymbol{I}),\boldsymbol{\xi} \sim \mathcal{N}(\boldsymbol{\mu},\boldsymbol{\sigma}^2\boldsymbol{I}),\boldsymbol{z}=f_{\boldsymbol{\theta}}(\boldsymbol{\epsilon}),\boldsymbol{y}=h_{\boldsymbol{\theta}}(\boldsymbol{\xi}),\boldsymbol{x}=G(\boldsymbol{z},\boldsymbol{y})}\Big[-\log |\det(\frac{\partial f_{\boldsymbol{\theta}}}{\partial \boldsymbol{\epsilon}})| - \log |\det(\frac{\partial h_{\boldsymbol{\theta}}}{\partial \boldsymbol{\xi}})|$$
$$- \log p_{\boldsymbol{z}}(\boldsymbol{z}) - \log p_{\boldsymbol{y}}(\boldsymbol{y}) + E_{\mathcal{C}}(\boldsymbol{x})\Big] - \mathbb{H}_{\mathcal{N}(\mathbf{0},\boldsymbol{I})} - \mathbb{H}_{\mathcal{N}(\boldsymbol{\mu},\boldsymbol{\sigma}^2\boldsymbol{I})} + \log Z.$$

---

**Algorithm 3:** Extension of Algorithm 2 to Conditional INN

**while** *not converged* **do**

    1. Sample $\boldsymbol{\epsilon} \sim \mathcal{N}(\mathbf{0}, \boldsymbol{I}), \boldsymbol{\rho} \sim p_{\boldsymbol{\rho}}(\boldsymbol{\rho})$

    2. Map $\boldsymbol{\epsilon}$ to latent code $\boldsymbol{z} = f_{\boldsymbol{\theta}}(\boldsymbol{\epsilon}, \boldsymbol{\rho})$

    3. Map $\boldsymbol{z}$ to an image $\boldsymbol{x} = G(\boldsymbol{z})$

    4. Optimize $\boldsymbol{\theta}$ with gradient $\nabla_{\boldsymbol{\theta}} \Big( -\log|\det(\frac{\partial f_{\boldsymbol{\theta}}}{\partial \boldsymbol{\epsilon}})| - \log p_{\boldsymbol{z}}(\boldsymbol{z}) + E_{\mathcal{C}_{\boldsymbol{\rho}}}(\boldsymbol{x}) \Big)$

---

## B.5 Extension to Conditional INN

Algorithm 3 illustrates how we extend the training algorithm (Algorithm 2) to conditional INN, which helps generalize to controls specified by continuous values $\boldsymbol{\rho}$. The training objective becomes

$$\arg\min_{\boldsymbol{\theta}} \mathbb{E}_{\boldsymbol{\rho} \sim p_{\boldsymbol{\rho}}(\boldsymbol{\rho})} \big[ \mathbb{D}_{\mathsf{KL}}\big( p_{\boldsymbol{\theta}}(\boldsymbol{z}|\boldsymbol{\rho}) \| p(\boldsymbol{z}|\mathcal{C}_{\boldsymbol{\rho}}) \big) \big], \tag{31}$$

where $\mathcal{C}_{\boldsymbol{\rho}}$ means that the control is specified by value $\boldsymbol{\rho}$. The derivation of Algorithm 3 is given by:

$$
\begin{aligned}
&\mathbb{E}_{\boldsymbol{\rho} \sim p_{\boldsymbol{\rho}}(\boldsymbol{\rho})} \big[ \mathbb{D}_{\mathsf{KL}}\big( p_{\boldsymbol{\theta}}(\boldsymbol{z}|\boldsymbol{\rho}) \| p(\boldsymbol{z}|\mathcal{C}_{\boldsymbol{\rho}}) \big) \big] \\
&= \mathbb{E}_{\boldsymbol{\rho} \sim p_{\boldsymbol{\rho}}(\boldsymbol{\rho}), \boldsymbol{z} \sim p_{\boldsymbol{\theta}}(\boldsymbol{z}|\boldsymbol{\rho})} \Big[ \log \frac{p_{\boldsymbol{\theta}}(\boldsymbol{z}|\boldsymbol{\rho})}{p(\boldsymbol{z}|\mathcal{C}_{\boldsymbol{\rho}})} \Big] \\
&= \mathbb{E}_{\boldsymbol{\rho} \sim p_{\boldsymbol{\rho}}(\boldsymbol{\rho}), \boldsymbol{z} \sim p_{\boldsymbol{\theta}}(\boldsymbol{z}|\boldsymbol{\rho}), \boldsymbol{x} = G(\boldsymbol{z})} \Big[ \log \frac{p_{\boldsymbol{\theta}}(\boldsymbol{z}|\boldsymbol{\rho})}{p_{\boldsymbol{z}}(\boldsymbol{z}) e^{-E_{\mathcal{C}_{\boldsymbol{\rho}}}(\boldsymbol{x})}/Z} \Big] \\
&= \mathbb{E}_{\boldsymbol{\rho} \sim p_{\boldsymbol{\rho}}(\boldsymbol{\rho}), \boldsymbol{\epsilon} \sim \mathcal{N}(\mathbf{0}, \boldsymbol{I}), \boldsymbol{z} = f_{\boldsymbol{\theta}}(\boldsymbol{\epsilon}, \boldsymbol{\rho}), \boldsymbol{x} = G(\boldsymbol{z})} \Big[ \log \mathcal{N}(\boldsymbol{\epsilon}|\mathbf{0}, \boldsymbol{I}) - \log|\det(\frac{\partial f_{\boldsymbol{\theta}}}{\partial \boldsymbol{\epsilon}})| \\
&\hspace{6cm} - \log p_{\boldsymbol{z}}(\boldsymbol{z}) + E_{\mathcal{C}_{\boldsymbol{\rho}}}(\boldsymbol{x}) + \log Z \Big] \\
&= \mathbb{E}_{\boldsymbol{\rho} \sim p_{\boldsymbol{\rho}}(\boldsymbol{\rho}), \boldsymbol{\epsilon} \sim \mathcal{N}(\mathbf{0}, \boldsymbol{I}), \boldsymbol{z} = f_{\boldsymbol{\theta}}(\boldsymbol{\epsilon}, \boldsymbol{\rho}), \boldsymbol{x} = G(\boldsymbol{z})} \Big[ -\log|\det(\frac{\partial f_{\boldsymbol{\theta}}}{\partial \boldsymbol{\epsilon}})| - \log p_{\boldsymbol{z}}(\boldsymbol{z}) \\
&\hspace{6cm} + E_{\mathcal{C}_{\boldsymbol{\rho}}}(\boldsymbol{x}) \Big] - \mathbb{H}_{\mathcal{N}(\mathbf{0}, \boldsymbol{I})} + \log Z.
\end{aligned}
\tag{32}
$$

## B.6 Derivation for Latent-Space Moment Constraint

Given a mapping $\boldsymbol{\gamma} : \mathcal{X} \to \mathbb{R}^K$, moment constraint [7, 37] defines the target distribution $p(\boldsymbol{x}|\mathcal{C})$ as

$$p(\boldsymbol{x}|\mathcal{C}) = \underbrace{\arg\min_{p(\boldsymbol{x}|\mathcal{C})} \mathbb{D}_{\mathsf{KL}}(p(\boldsymbol{x}|\mathcal{C}) \| p_{\boldsymbol{x}}(\boldsymbol{x}))}_{\text{Deviation from the pre-trained distribution}}, \quad \text{s.t.} \quad \underbrace{\mathbb{E}_{\boldsymbol{x} \sim p(\boldsymbol{x}|\mathcal{C})} \big[ \boldsymbol{\gamma}(\boldsymbol{x}) \big] = \boldsymbol{\mu}}_{\text{Moment constraint}}, \tag{33}$$

where $\boldsymbol{\mu}$ is the user-specified constraint. Examples are provided in the main text, and we omit them here for brevity. [7] showed that Eq. (33) can be approximated to an arbitrary precision by

$$p(\boldsymbol{x}|\mathcal{C}) \propto p_{\boldsymbol{x}}(\boldsymbol{x}) \exp\big( \hat{\boldsymbol{\beta}}^\top \boldsymbol{\gamma}(\boldsymbol{x}) \big), \tag{34}$$

where $\hat{\boldsymbol{\beta}}$ needs to be computed. [37] estimates $\hat{\boldsymbol{\beta}}$ by solving the following regression problem:

$$\hat{\boldsymbol{\beta}} = \arg\min_{\boldsymbol{\beta}} \mathbb{E}_{\boldsymbol{x}^{(1)}, \ldots, \boldsymbol{x}^{(N)} \overset{\text{i.i.d.}}{\sim} p_{\boldsymbol{x}}(\boldsymbol{x})} \left\| \frac{\sum_{j=1}^N \exp\big( \boldsymbol{\beta}^\top \boldsymbol{\gamma}(\boldsymbol{x}^{(j)}) \big) \boldsymbol{\gamma}(\boldsymbol{x}^{(j)})}{\sum_{j'=1}^N \exp\big( \boldsymbol{\beta}^\top \boldsymbol{\gamma}(\boldsymbol{x}^{(j')}) \big)} - \boldsymbol{\mu} \right\|_2^2. \tag{35}$$

In [37], sampling $\boldsymbol{x} \sim p_{\boldsymbol{x}}(\boldsymbol{x})$ is straightforward since they focus on autoregressive language models. In the context of latent-variable vision generative models, where $\boldsymbol{x} \sim p_{\boldsymbol{x}}(\boldsymbol{x})$ is implicitly defined as $\underline{\boldsymbol{z} \sim p_{\boldsymbol{z}}(\boldsymbol{z}), \boldsymbol{x} = G(\boldsymbol{z})}$, Eq. (35) is equivalent to

$$\hat{\boldsymbol{\beta}} = \arg\min_{\boldsymbol{\beta}} \mathbb{E}_{\boldsymbol{z}^{(1)}, \ldots, \boldsymbol{z}^{(N)} \overset{\text{i.i.d.}}{\sim} p_{\boldsymbol{z}}(\boldsymbol{z}), \boldsymbol{x}^{(j)} = G(\boldsymbol{z}^{(j)})} \left\| \frac{\sum_{j=1}^N \exp\big( \boldsymbol{\beta}^\top \boldsymbol{\gamma}(\boldsymbol{x}^{(j)}) \big) \boldsymbol{\gamma}(\boldsymbol{x}^{(j)})}{\sum_{j'=1}^N \exp\big( \boldsymbol{\beta}^\top \boldsymbol{\gamma}(\boldsymbol{x}^{(j')}) \big)} - \boldsymbol{\mu} \right\|_2^2. \tag{36}$$

Finally, based on Proposition 1, we can generalize the moment constraint to the latent space as

$$p(\boldsymbol{z}|\mathcal{C}) = \frac{p_{\boldsymbol{z}}(\boldsymbol{z})\exp\left(\hat{\boldsymbol{\beta}}^{\top}\boldsymbol{\gamma}\big(G(\boldsymbol{z})\big)\right)}{Z}, \quad Z = \int_{\boldsymbol{z}'} p_{\boldsymbol{z}}(\boldsymbol{z}')\exp\left(\hat{\boldsymbol{\beta}}^{\top}\boldsymbol{\gamma}\big(G(\boldsymbol{z}')\big)\right)d\boldsymbol{z}'. \quad (37)$$

Note that Eq. (36) and Eq. (37) are verbatim copies of Eq. (7) and Eq. (6), respectively.

---

**Algorithm 4:** Extension of Algorithm 3 with ID Energy (Section 4.3)

---

**while** *not converged* **do**

    1. Sample $\boldsymbol{\epsilon} \sim \mathcal{N}(\mathbf{0}, \boldsymbol{I}), \boldsymbol{\rho} \sim p_{\boldsymbol{\rho}}(\boldsymbol{\rho})$
    2. Map $\boldsymbol{\epsilon}$ to latent codes $\boldsymbol{z} = f_{\boldsymbol{\theta}}(\boldsymbol{\epsilon}, \boldsymbol{\rho})$ and $\boldsymbol{z}_0 = f_{\boldsymbol{\theta}}(\boldsymbol{\epsilon}, \boldsymbol{\rho}_0)$
    3. Map $\boldsymbol{z}$ to an image $\boldsymbol{x} = G(\boldsymbol{z})$ and $\boldsymbol{z}_0$ to an image $\boldsymbol{x}_0 = G(\boldsymbol{z}_0)$
    4. Optimize $\boldsymbol{\theta}$ with gradient

$$\nabla_{\boldsymbol{\theta}}\Big(-\log|\det(\frac{\partial f_{\boldsymbol{\theta}}}{\partial \boldsymbol{\epsilon}})| - \log p_{\boldsymbol{z}}(\boldsymbol{z}) + E_{\mathcal{C}_{\boldsymbol{\rho}}}(\boldsymbol{x}) + \lambda_{\mathrm{ID}} E_{\mathrm{ID}}(\boldsymbol{x}_0, \boldsymbol{x})\Big)$$

---

### B.7 Inverse Graphics Control with Identity Energy

This subsection provides more detail on how we use an inverse graphics model, DECA [16], to control the pose of faces generated by StyleGAN2 trained on FFHQ $1024^2$, as introduced in Section 4.3. Recall that we use the conditional INN extension, detailed in Algorithm 3, for this experiment. To enable generating different poses of the same identity, we add additional identity energy using the IR-SE50 model [8]. The training algorithm is detailed in Algorithm 4. Specifically, we generate a canonical latent code $\boldsymbol{z}_0 = f_{\boldsymbol{\theta}}(\boldsymbol{\epsilon}, \boldsymbol{\rho}_0)$ for each $\boldsymbol{\epsilon}$, where $\boldsymbol{\rho}_0$ is the canonical pose. The latent code $\boldsymbol{z}$ and the canonical latent code $\boldsymbol{z}_0$ are mapped to the image $\boldsymbol{x}$ and the canonical image $\boldsymbol{x}_0$. The identity energy in Eq. (12), copied below, encourages the embeddings of the two images to be similar:

$$E_{\mathrm{ID}}(\boldsymbol{x}_0, \boldsymbol{x}) = 1 - \cos\langle R(\boldsymbol{x}_0), R(\boldsymbol{x})\rangle, \quad \boldsymbol{x}_0 = G(\boldsymbol{z}_0), \boldsymbol{x} = G(\boldsymbol{z}). \quad (38)$$

### B.8 Experimental Details

Our INN architecture contains 8 blocks. Each block consists of a soft permutation of channels [2], an affine coupling layer [12], and an ActNorm layer [39]. In the affine coupling layer, we model the sub-network as an MLP with one hidden layer, where the hidden dimension is 256 and the non-linear activation is LeakyReLU$_{0.1}$.

For the prior distribution $p_{\boldsymbol{\rho}}(\boldsymbol{\rho})$ of the pose parameter $\boldsymbol{\rho}$ in the inverse graphics experiments (Section 4.3), we sample the $x$-axis and $y$-axis rotations relative to a canonical pose $\boldsymbol{\rho}_0$, whose $x$-axis and $x$-axis rotations are 0.3 and 0, respectively. The relative pose's $x$-axis and $y$-axis rotations are uniformly sampled from $[-(1/9)\pi, (1/9)\pi]$.

Each experiment was run on an NVIDIA RTX A4000 GPU (with 16G memory). Our code implementation is based on the PyTorch framework. Our code can be found at `https://github.com/ChenWu98/Generative-Visual-Prompt`.

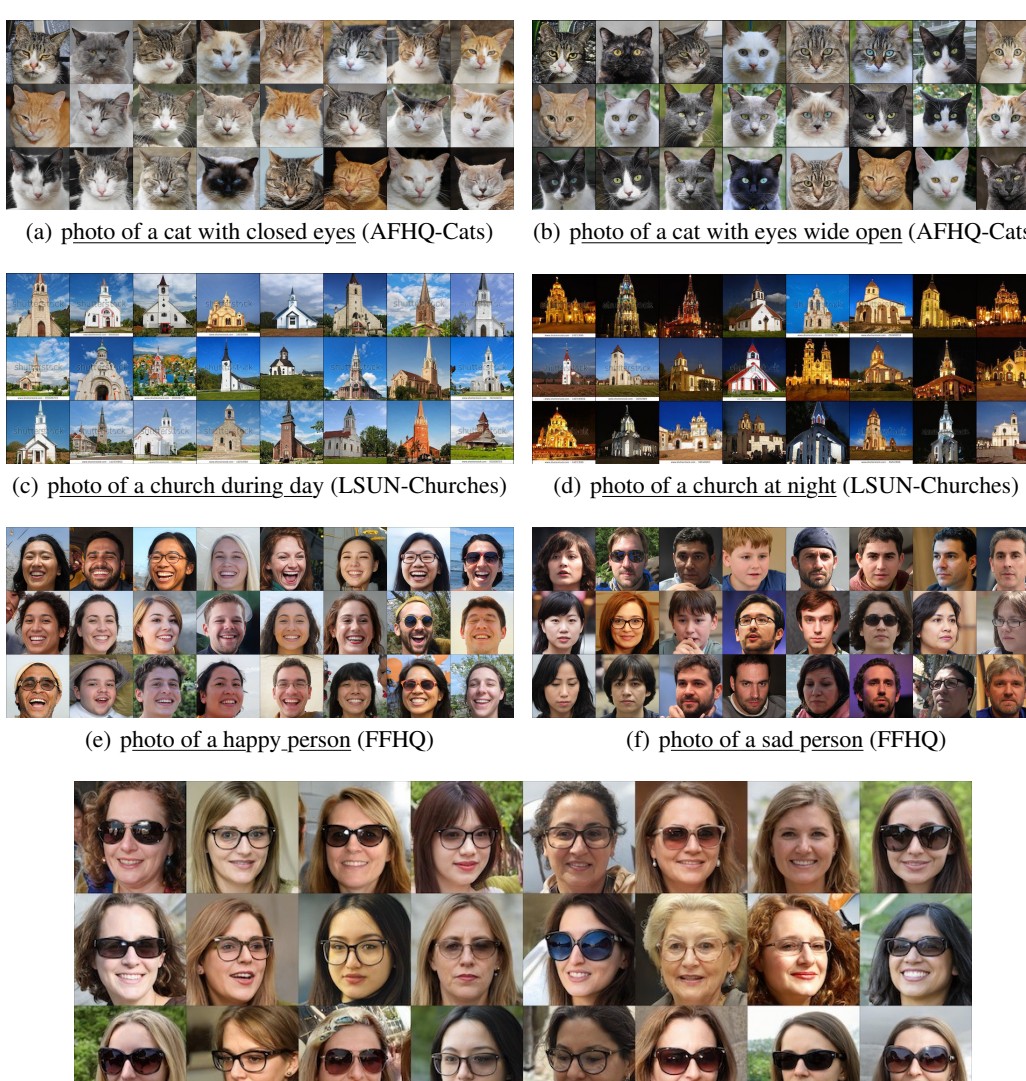

(a) photo of a cat with closed eyes (AFHQ-Cats)

(b) photo of a cat with eyes wide open (AFHQ-Cats)

(c) photo of a church during day (LSUN-Churches)

(d) photo of a church at night (LSUN-Churches)

(e) photo of a happy person (FFHQ)

(f) photo of a sad person (FFHQ)

(g) photo of a woman with glasses (FFHQ)

Figure 10: Additional results for image synthesis based on text description, guided by the CLIP model. As the pre-trained generative model, we use StyleGAN2 trained on FFHQ, AFHQ-Cats and LSUN-Churches datasets. The captions are the text descriptions given to the CLIP model.

## C   Additional Results for Image Synthesis based on Text Description

Figure 10 presents additional results of PromptGen for image synthesis from text, using the CLIP model as control. As the pre-trained generative model, we explore StyleGAN2 trained on AFHQ-Cats [6], LSUN-Churches [83], and FFHQ [35].

## D   Additional Results for De-Biasing Generative Models

Table 5 provides quantitative results of de-biasing StyleGAN2 across categorical attributes (Figure 6). Specifically, we used the pre-trained classifier provided by FairFace [32] to classify the attributes of the generated images. We then report the KL divergence between the attribute distribution of the generated images and the uniform distribution. Figure 11 and Figure 12 visualize the de-biasing results. We report the KL divergence between the generated distribution and the uniform distribution.

Table 5: Quantitative results for de-biasing categorical attributes (Figure 6). See details in Appendix D. PromptGen de-biases StyleGAN2 in terms of race, age, and gender.

| | FFHQ | | MetFaces | | |
|---|---|---|---|---|---|
| | $\mathbb{D}_{KL}^{race}\downarrow$ | $\mathbb{D}_{KL}^{age}\downarrow$ | $\mathbb{D}_{KL}^{race}\downarrow$ | $\mathbb{D}_{KL}^{age}\downarrow$ | $\mathbb{D}_{KL}^{gender}\downarrow$ |
| StyleGAN2 | 0.860 | 0.597 | 1.624 | 0.546 | 0.019 |
| PromptGen (ours; $\lambda = 1$) | 0.286 | 0.357 | 0.687 | 0.397 | **0.000** |
| PromptGen (ours; $\lambda = 2$) | **0.099** | **0.172** | **0.189** | **0.247** | 0.005 |

Interestingly, on MetFaces, de-biasing the race results in more sculptures, and we postulate that the reason is that almost all paintings and sketches in MetFaces are for white individuals.

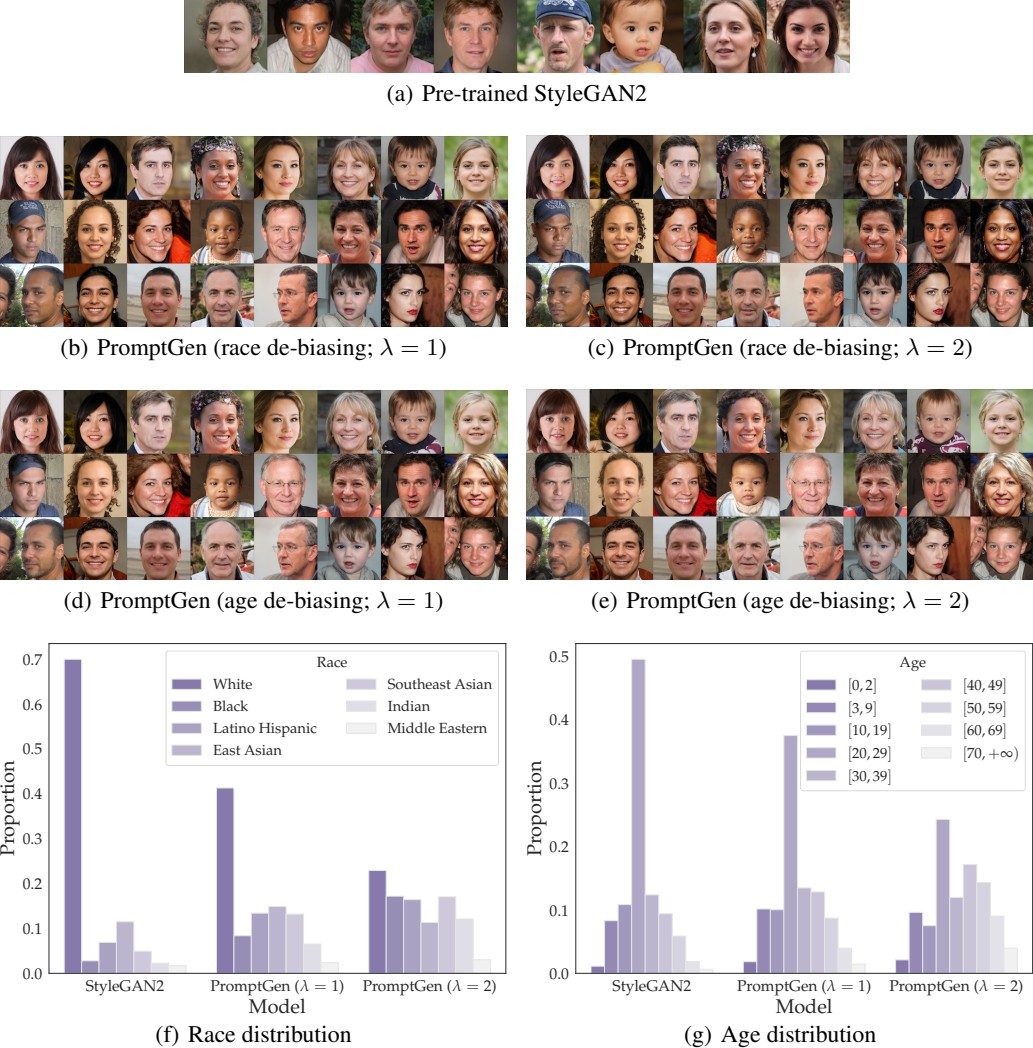

(a) Pre-trained StyleGAN2

(b) PromptGen (race de-biasing; $\lambda = 1$)

(c) PromptGen (race de-biasing; $\lambda = 2$)

(d) PromptGen (age de-biasing; $\lambda = 1$)

(e) PromptGen (age de-biasing; $\lambda = 2$)

(f) Race distribution

(g) Age distribution

Figure 11: Using our moment constraint, PromptGen de-biases the racial and age distributions of StyleGAN2 trained on FFHQ with truncation $\psi = 0.7$. All synthesized images are $1024^2$ in resolution and resized for visualization. We fixed the random seed for PromptGen, so please zoom in to see detailed differences between images.

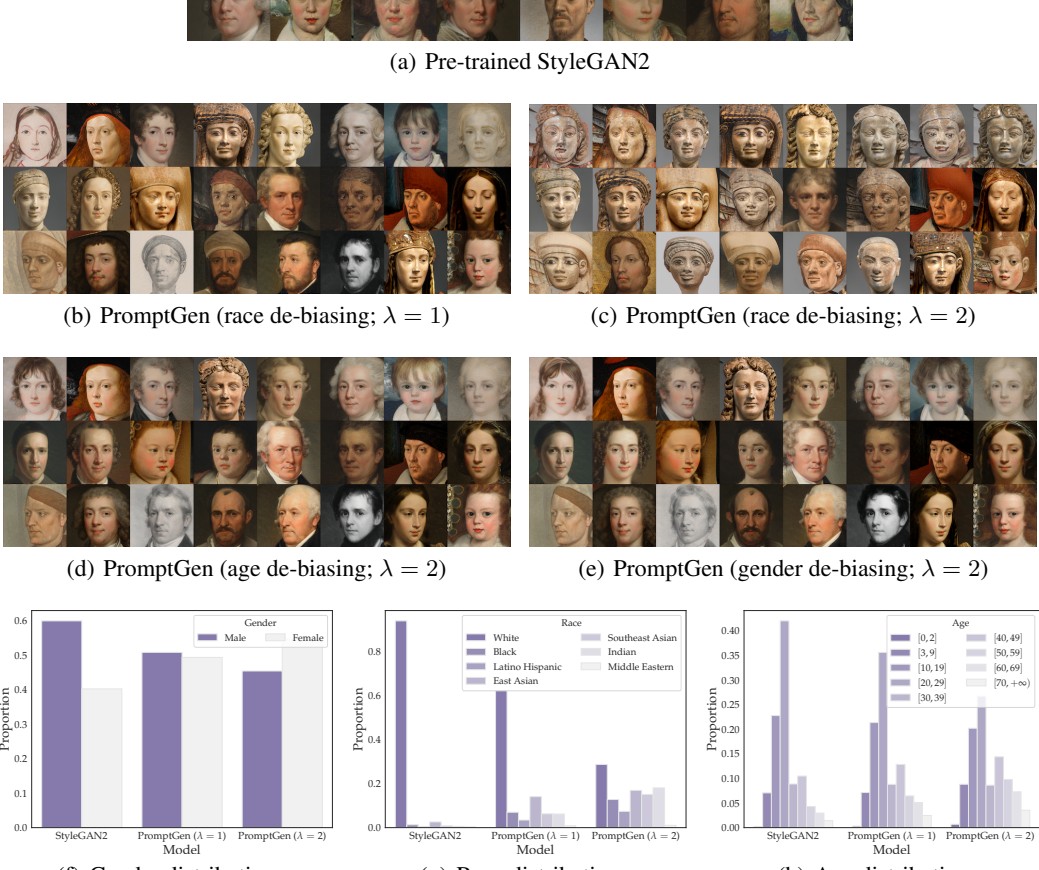

(a) Pre-trained StyleGAN2

(b) PromptGen (race de-biasing; $\lambda = 1$)   (c) PromptGen (race de-biasing; $\lambda = 2$)

(d) PromptGen (age de-biasing; $\lambda = 2$)   (e) PromptGen (gender de-biasing; $\lambda = 2$)

(f) Gender distribution   (g) Race distribution   (h) Age distribution

Figure 12: Using the moment constraint, PromptGen de-biases the racial, age, and gender distributions of StyleGAN2 trained on MetFaces with truncation $\psi = 0.7$. All images are $1024^2$ and resized for visualization. Interestingly, de-biasing the race results in more sculptures, and we postulate that the reason is that almost all paintings and sketches in MetFaces are for white individuals. We fixed the random seed for PromptGen, so we recommend zooming in to see differences between images.

### D.1 Decomposing Complex Control via Energy Composition

Energy composition $E_{\mathcal{C}}(\boldsymbol{x}) = \sum_{i=1}^{M} \lambda_i E_i(\boldsymbol{x}, \boldsymbol{y}_i)$ in Eq. 1 allows us to decompose a complex controls into simple ones. For instance, Figure 13(a) shows that $\boldsymbol{y}_1 = \underline{\text{photo of a bald black man with beard}}$ ($\lambda_1 = 6000$) generates good samples of men with beard but several of them have light-skin tone; however, by decomposing it as $\boldsymbol{y}_1 = \underline{\text{photo of a bald man}}$ ($\lambda_1 = 1500$), $\boldsymbol{y}_2 = \underline{\text{photo of a black man}}$ ($\lambda_1 = 3000$), and $\boldsymbol{y}_3 = \underline{\text{photo of a man with beard}}$ ($\lambda_1 = 1500$), we have (on average) darker skin, as shown in Figure 13(b). Note that the random seed for Figure 13(a) and Figure 13(b) is the same, and hence the identities are preserved and mostly the skin tone has changed.

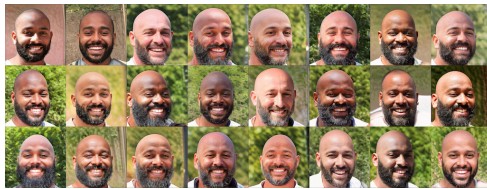 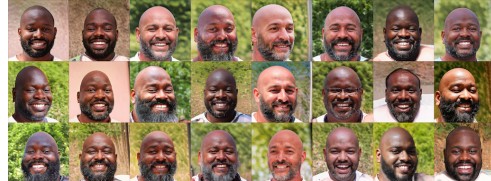

(a) CLIP with one complex sentence  (b) CLIP with three simple sentences

Figure 13: Decomposing complex controls (e.g., photo of a bald black man with beard) into simpler ones improves control performance. See details in Section D.1.

## E  Experiments on Generative 3D Face Models

3D face modeling has been an active area of research that has recently gained major interest due to applications in virtual humans, deep faces, and digital actors. Existing 3D deep learning generative models build 3D compact representations of shape and appearance capable of modeling non-linearity (e.g., scatter effects, specularities) that is necessary for generating photo-realistic faces. Providing a generative model with the capability of generating 3D faces with a particular geometry (e.g., a specific distance between eyes or nose length) is particularly useful for technical artists when creating new characters. However, this problem remains unaddressed, partially due to the lack of publicly available 3D databases labeled with such constraints. We refer to this capability as the intra-sample constraint in 3D generative models, and this section shows how our PromptGen framework is able to achieve this by defining a new energy function based on signed distances.

We conducted the experiments on the FaceScape dataset [82], a large-scale 3D human face dataset consisting of $16,940$ topologically uniformly registered 3D face meshes along with high-quality textures. The dataset contains $847$ identities with $20$ expressions per identity, and each mesh consists of $26,317$ vertices of which $68$ are 3D landmark vertices covering eyes, nose, mouth, jaw, and eyebrows. With some abuse of notation, we denote a 3D face mesh as $\boldsymbol{x} \in \mathbb{R}^{3V}$ (recall that $\boldsymbol{x}$ is used to represent an image in other parts of this paper), where $V = 26,317$ is the number of vertices. Given an index $l$ of a vertex (e.g., one of the 68 landmark vertices), the 3D coordinate of this vertex is defined as $\boldsymbol{x}[l] \in \mathbb{R}^3$. We pre-train a generative mesh model $G$ that maps each latent code $\boldsymbol{z} \sim \mathcal{N}(\boldsymbol{0}, \boldsymbol{I})$ to a 3D mesh $\boldsymbol{x}$. PromptGen allows us to obtain geometric control on the 3D meshes. Specifically, given two landmark vertex indices $l_1$ and $l_2$, we define an intra-sample constraint that enforces the signed distance (along a given direction) between the two vertices to be $s$. To this effect, we define the signed-distance energy as

$$E_{\text{signed-distance}}(\boldsymbol{x}) = |d(\boldsymbol{v}_1, \boldsymbol{v}_2) - s|, \quad \boldsymbol{v}_1 = \boldsymbol{x}[l_1], \boldsymbol{v}_2 = \boldsymbol{x}[l_2] \tag{39}$$

where $d(\cdot, \cdot)$ is the signed-distance (along a given direction), and $|\cdot|$ is the absolute value.

Figure 14 presents several examples of 3D mesh control, in which random textures are applied to the generated meshes. It shows that PromptGen successfully controls the generative 3D mesh model in terms of the mouth stretch, eye openness, nose-tip stretch, and lip forward.

## F  Error Analysis

As we discussed in Section 5, the controlled distribution depends on (1) the pre-trained generative model's coverage and (2) the off-the-shelf models used for the control. In this section, we provide

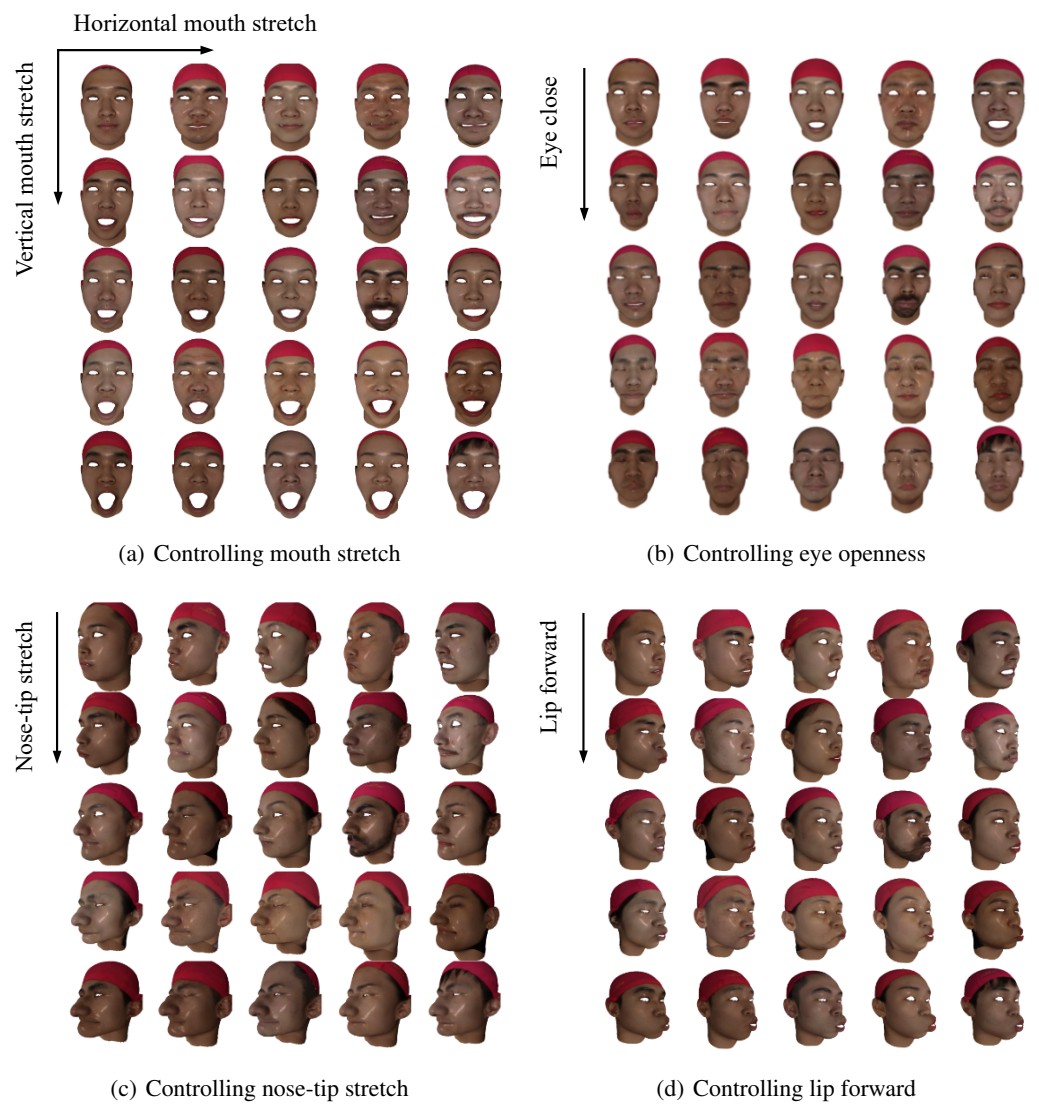

(a) Controlling mouth stretch

(b) Controlling eye openness

(c) Controlling nose-tip stretch

(d) Controlling lip forward

Figure 14: PromptGen with intra-sample signed-distance constraints that generate 3D face models with varying distances in the mouth, eye openness, nose-tip stretch, and lip forward.

some examples of PromptGen failing to generate satisfactory images using the CLIP energy. For the pre-trained generative model, we use StyleGAN2 trained on FFHQ, AFHQ-Wild, or LSUN-Churches.

Our first observation is that the CLIP model sometimes fails in modeling linguistic negation. For instance, the text description photo of a man without beard results in a distribution of photos of men with beard (Figure 15(a)). Meanwhile, Figure 15(b) shows that CLIP is capable of modeling the negation of having makeup, but with the "reporting bias" discussed in Section 4.4. Moreover, CLIP seems to have difficulty in numerical reasoning, and gaining control over the count of specific objects in a scene tends to be unsuccessful. We showed this by specifying photo of a church with three windows, which did not result in the desired specification (Figure 15(c)).

To gain a deeper understanding of the above failures and biases, in Figure 16, we provide some image samples from the LAION-400M dataset [65] with CLIP retrieval, using the same text descriptions as Figure 15. CLIP retrieval is based on the CLIP embedding similarity between the web images and text descriptions, while the original text below each individual image is not used. We observe an impressive consistency between CLIP retrieval and PromptGen: in both Figure 15(a) and Figure 16(a), most images have beard; in both Figure 15(b) and Figure 16(b), all images are female; in both Figure 15(c) and Figure 16(c), some images do not have exactly three windows. This consistency suggests that

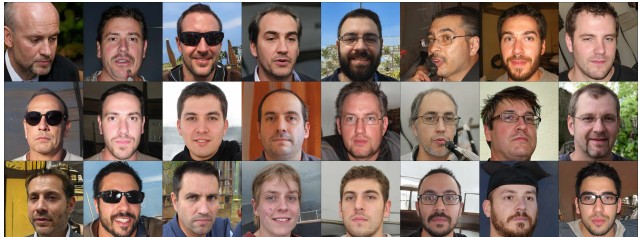

(a) photo of a man without beard (FFHQ)

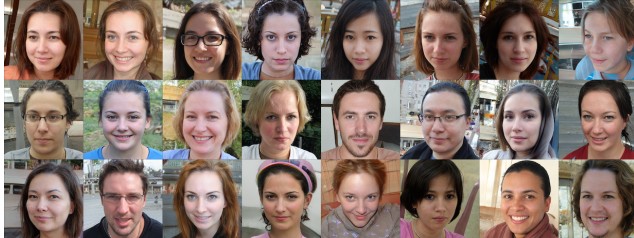

(b) a photo of a person without makeup (FFHQ)

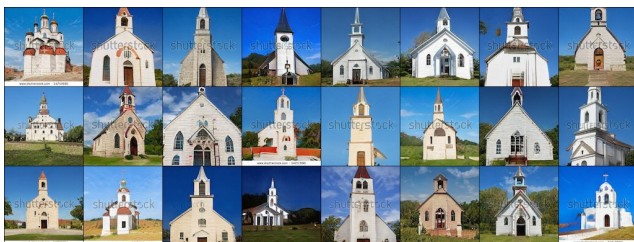

(c) photo of a church with three windows (LSUN-Churches)

Figure 15: When the text description contains certain linguistic properties (e.g., negation, numerical reasoning), CLIP sometimes fails or shows the "reporting bias" that we discuss in Section 4.4. For a deeper understanding of these failures and biases, in Figure 16, we provide some image samples from the LAION-400M dataset [65] with CLIP retrieval, using the same text descriptions as this figure.

the failures and biases in Figure 15 should be mostly attributed to the CLIP model rather than to our PromptGen algorithm. We believe that our observation sheds light on the intricacy of contrastive multi-modal (vision-language) pre-training, which is worthy of being further investigated.

Another observation is that the control tends to fail when the text description requires sampling from low-density regions of the pre-trained generative model's output space. In other words, the control usually fails if the pre-trained generative model does not cover the mode we are trying to gain control over. For example, images faithful to photo of a person yawning and photo of a baby with long hair are not commonly observed in the FFHQ dataset and, hence, these two text descriptions result in degeneration (Figure 17(a)) or weird images (Figure 17(b)). Another example is photo of an animal from the side, which is not commonly observed in the AFHQ-Wild dataset, and Figure 17(c) shows that the generated images fail to follow this description. Even when the control is successful (e.g., when a complex description is decomposed in Figure 13(b)), sampling from low-density regions results in limited diversity (e.g., the backgrounds in Figure 13(b) look similar to each other).

Finally, we also ran some failure controls using PPGM, showing that PromptGen and PPGM reveal similar failure cases of pre-trained generative models and CLIP. Results are shown in Figure 18.

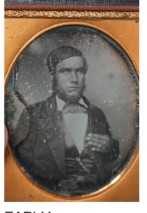 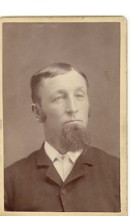 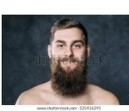 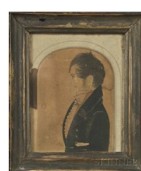 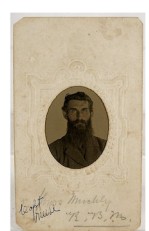 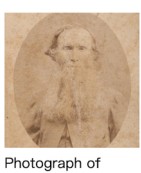 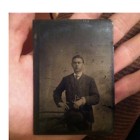

EARLY PHOTOGRAPHY LOT WITH 2 DAGUERREOTYPES & 4 – ...

Brystbilde av Halvor N. Flåta

Caucasian man with funny spinning mustache and big...

Miniature Portrait, watercolor and ink on paper, c...

33rd Virginia Infantry – Captain Muse of Company D...

Photograph of Charles Harpur

Tintype of young gentleman

(a) photo of a man without beard (CLIP retrieval from LAION-400M)

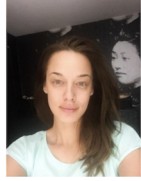 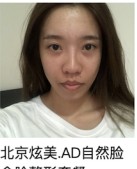 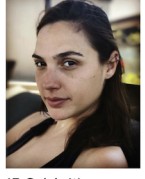 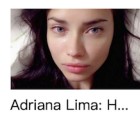 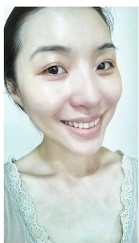 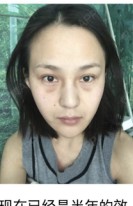 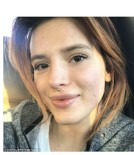

Andreea Raicu

北京炫美.AD自然脸全脸整形套餐

15 Celebrities without Makeup Prove They Look Just...

Adriana Lima: H... κουρασμένη selfie της

004混合偏油性肌膚

现在已经是半年的效果感觉脸还是瘦瘦的效果挺好的医生手法

Glowing: Bella, 20, showed off her complexion on l...

(b) a photo of a person without makeup (CLIP retrieval from LAION-400M)

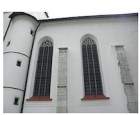 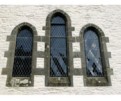 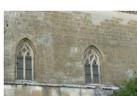 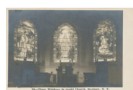 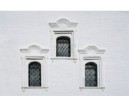 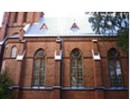 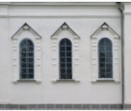

St. Procopius Church, Žďár nad Sázavou – The later...

γοτθικά Windows Στοκ φωτογραφίες με δικαίωμα ελεύθ...

Detail of the church exterior wall showing numerou...

Roxbury NY *3 Stained Glass Windows in Gould Churc...

Oude vensters op witte muur Royalty–vrije Stock Fo...

Tartu katoliku kirik – vaade Veski tänavalt.jpg

Церковь Рождества Христова – Клястицкое –

(c) photo of a church with three windows (CLIP retrieval from LAION-400M)

Figure 16: For a deeper understanding of the failures and biases illustrated in Figure 15, we provide some image samples from the LAION-400M dataset [65] with CLIP retrieval, using the same text descriptions as Figure 15. CLIP retrieval is based on the CLIP embedding similarity between the web images and text descriptions, while the original text below each individual image is not used. In Figure 16(a), most images have a beard; in Figure 16(b), all images are female; in Figure 16(c), some images do not have exactly three windows. These observations are consistent with those in Figure 15.

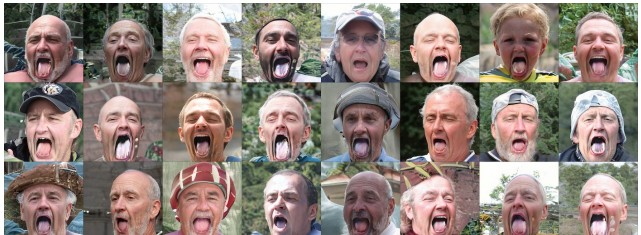

(a) photo of a person yawning (FFHQ)

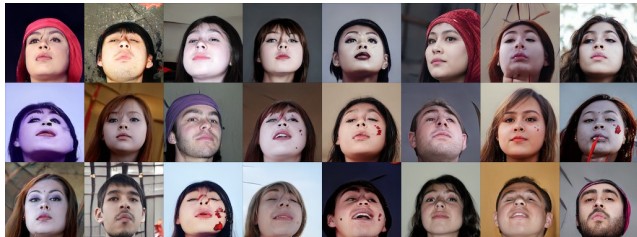

(b) photo of a baby with long hair (FFHQ)

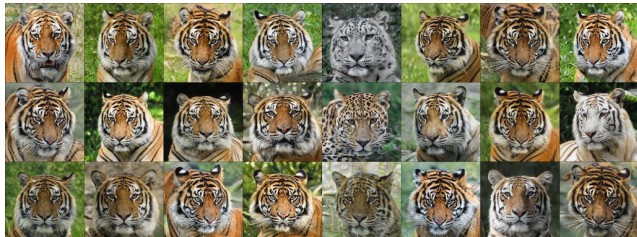

(c) photo of an animal from the side (AFHQ-Wild)

Figure 17: When the pre-trained generative model fails in covering certain modes required by the text description, unsatisfactory outputs are produced. In this figure, we show several text descriptions that require sampling from low-density regions of the pre-trained generative model's output space.

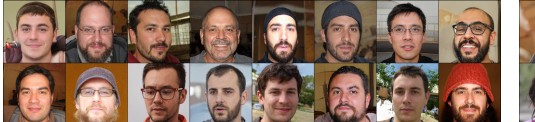
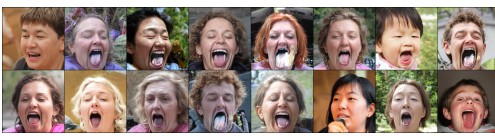

(a) photo of a man without beard (FFHQ; w/ PPGM)    (b) photo of a person yawning (FFHQ; w/ PPGM)

Figure 18: Failure modes revealed by PromptGen (Figure 15 and Figure 17) also hold for PPGM.