# OpenReview forum: "Generative Visual Prompt: Unifying Distributional Control of Pre-Trained Generative Models"
_NeurIPS.cc/2022/Conference — NeurIPS 2022 Accept_

### Official Review · Reviewer_QXTM · 2022-07-09

**Rating:** 5
**Confidence:** 4
**Soundness:** 3 good
**Presentation:** 3 good
**Contribution:** 3 good

**Summary:**

This work proposes a unified method called PromptGen to control the image synthesis of pre-trained generative models, such as StyleGAN, NVAE and StyleNeRF. The basic idea is to 1) first formulate the latent variable z distribution conditioned on the control \mathcal{C}, i.e., $p(x|\mathcal{C})$ as a latent-space EBM, where the latent-space EBM can use pre-trained image classifiers, CLIP model and inverse graphics model to specify the control; and 2) train an invertible neural network (INN) to approximate the latent-space EBM using an KL divergence as the training objective. In experiments across different image datasets, this work shows the efficacy and efficiency of PromptGen in the tasks of image synthesis based on text description, de-biasing generative models, pose-guided face synthesis, and iterative control via functional composition.


**Questions:**

(1) Can more quantitative results on the controllability be added to show the performance of PromptGen?

(2) In Figure 5(d), I think we see a severe mode collapse issue, as the background colors and object patterns are very similar across different generated images. How do you explain this phenomenon?

(3) For the text-guided generation, can PromptGen perform well with more complex sentences, where we have a composition of multiple attributes, such as “a photo of a smiling baby with glasses”? These kinds of results will better demonstrate the compositionality of the proposed method.

(4) For the generative models with a class-embedding space, I wonder why we need to train another latent-space generator $y=h_\theta(\xi)$? I feel like we can also directly train a single latent-space generation $z=f_\theta(\epsilon, y)$ to approximate the EBM. Any justification?


**Limitations:**

This work has well addressed their limitations. But I didn’t see many discussions about their negative societal impact. I think this work shares with other image synthesis tools similar potential benefits and risks, which have been discussed extensively in [1]. I suggest adding more discussions about the risks of controllable image synthesis.

[1] Vaccari, C. and Chadwick, A. (2020). Deepfakes and disinformation: Exploring the impact of synthetic political video on deception, uncertainty, and trust in news. Social Media+ Society.


**Strengths And Weaknesses:**

Strengths:

(1) Controlling generative models using EBM with MCMC Langevin dynamics is effective but slow (it needs multiple inference/optimization iterations). Thus, this work proposes to train another latent generator $z=f_\theta(\epsilon)$ that approximates the latent-space EBM, so the sampling can be performed with one forward pass, which I think is the main originality and significance of this work.

(2) For the second contribution, with the latent generator $z=f_\theta(\epsilon)$, this work also demonstrates 1) the generality for different controllable generation tasks, and 2) the iterative controls by composing the latent generators and the pre-trained generative model.

(3) The paper is well-written and easy to read. Extensive experiments were performed to show the wide applicability, effectiveness and efficiency of the proposed method.

Weaknesses:

My main concern is that in experiments, this work didn’t provide sufficient quantitative results of evaluating the controllability. I like the quantitative results of evaluating the de-biasing performance, but regarding other controllable generation tasks, such as text-guided generation and pose-guided generation, I didn’t see the quantitative results of how the proposed method controls the generation to satisfy the specified attribute or text.

For other concerns, please see the questions below.

---

> ### Author Response · Authors · 2022-08-01
> **Author response**
>
> **Quantitative analysis for text and pose controls**
>
> In Section 4.6 (Section 4.5 in the original version), we reported quantitative results for controllability and inference speed in the text control setting. For the pose control, we visually verified that for different people the pose is consistent.
>
> **Lack of diversity on ImageNet in Figure 5(e) [previously 5(d)]**
>
> First, we refer to **Clarification: PromptGen in the class-embedding space** section in **General response to all reviewers**. The ImageNet experiment is mainly designed to show that PromptGen can not only model distributions in the $\boldsymbol{z}$-space, but also the class-embedding space, an extension of our main method. However, we agree that this extension suffers from low diversity; in the updated version, we mentioned this limitation.
>
>
> **More discussion on compositionality and complex control**
>
> Thanks for pointing it out! First, we refer to **Clarification: energy composition v.s. functional composition** section in **General response to all reviewers**. What you mentioned is what we call energy composition, and we added a new section (Section 4.5) to discuss this issue. Specifically, we show that control with a complex sentence (i.e.,*a photo of a bald black man with beard*) is not very successful. We then decompose the complex sentence into three simpler ones (i.e., *a photo of a black man*, *a photo of a bald man*, and *a photo of a man with beard*), and we find that by tuning the weight $\lambda_i$ for each of them (e.g., we need larger $\lambda_i$ for *a photo of a black man*), we have better control performance.
>
>
> **PromptGen for generative models with a class embedding space**
>
> First, we refer to **Clarification: PromptGen in the class-embedding space** section in **General response to all reviewers**. The setting you mention in question (4) is what we refer to as *PromptGen in the $\boldsymbol{z}$-space of a class-conditional GAN*. As you mentioned, this setting is the same as the unconditional setting discussed throughout this paper, given the fact that $G(\cdot, \text{Emb}(c))$ has the same form as an unconditional generative model.
>
> On the other hand, *PromptGen in the class-embedding space of a class-conditional GAN* aims at learning a distribution in the class-embedding space. This setting exploits the observation that a class-conditional GAN has a certain ability to generate out-of-domain samples when the provided class embedding is not one of the classes in ImageNet. Under this setting, PromptGen aims to find a distribution over class embeddings $y$ such that $G(z, y)$ are “photos of a glow and light dog”. However, results in Figure 5(e) [previously 5(d)] show that the diversity of images sampled from such control is limited; in the updated version, we mentioned this limitation.

---

> > ### Comment · Reviewer_QXTM · 2022-08-08
> > **My major concerns still remain**
> >
> > I thank the authors for their response. However, (1) my major concerns on “this work didn’t provide sufficient quantitative results of evaluating the controllability” have not been addressed. The authors mentioned that the CLIP energy has been used for measuring the controlling performance of text-guided generation. But the CLIP energy is the training or inference objective for the PPGM and PromptGen methods, which, I think, should not be an appropriate evaluation metric any more. (2) Also, it seems that the proposed method does not work well with complex text sentences in the text control case, but needs many hand-crafted heuristics to compose energy functions with careful hyperparameter tunings. This makes me less convinced of its effectiveness in text-guided generation. (3) Since the setting of “PromptGen in the class-embedding space of a class-conditional GAN” suffers from a severe diversity issue, it seems to be better to present it as a failure case.
> >
> > Thus, I lowered my score to “borderline accept”.

---

> > > ### Author Response · Authors · 2022-08-09
> > > **Author response**
> > >
> > > **Is CLIP energy an appropriate evaluation metric?**
> > >
> > > Thanks for raising this question! We would like to argue that CLIP energy is an adequate evaluation metric; using the same off-the-shelf model (or models trained on the same dataset) for optimization and evaluation has been adopted in previous works. In PPGM [47], Table S3 used the same image classifier for optimization and evaluation. In LACE [50], Tables 1 and 2 used a latent-space classifier for optimization and an image-space classifier for evaluation (the two classifiers are trained on the same dataset). Our usage of CLIP energy shares the same spirit as their usage of classifier-based metrics.
> > >
> > >
> > > **Complex controls do not always work**
> > >
> > > We provided a comprehensive error analysis in Appendix F. We agree that complex CLIP sentences cause undesired effects in the generated images, but we have analyzed that many failure cases are caused by the low density in the training data of generative models. For instance: “Photo of a happy Asian person with a hat and glasses” performs worse than “Photo of a happy man with a hat and glasses” where “man” replaces “Asian person”. The complexity of these sentences in terms of the number of attributes they specify is the same however since there are more men than Asian people in the training set, the results are better. We have been preparing a more comprehensive set of experiments around this which we plan to include in the appendix. Regarding the hyperparameter tuning of the energy functions, we allow tuning the importance of decomposed descriptions to overcome the language modeling limitation of CLIP by adjusting only a weight parameter.
> > >
> > > Moreover, we would like to point out that, compared with previous papers on controlling generative models with latent-space EBMs (e.g., PPGM [47], LACE [50]) – which only consider classifier control – we experiment with more complex controls. Specifically, we have experiments on (1) CLIP guidance, (2) inverse graphics guidance, and (3) moments constraints. For (1), we re-implemented PPGM for comparison, while we did not re-implement LACE because we do not have CLIP models in the w-space of GANs (note that LACE used classifiers trained in the w-space instead of the image space). We did not re-implement PPGM and LACE for other guidance because (2) needs to train a network conditioned on poses, which we do not know how to apply to PPGM and LACE, and (3) is a core part of our methodology.
> > >
> > >
> > > **Diversity of PromptGen in the class-embedding space**
> > >
> > > We agree with the limitations of PromptGen in the *PromptGen in the class-embedding space* setting. We will clarify and emphasize this limitation in the final version of the paper.
> > >
> > >
> > >
> > > [47] Anh Nguyen, Jeff Clune, Yoshua Bengio, Alexey Dosovitskiy, and Jason Yosinski. Plug & play generative networks: Conditional iterative generation of images in latent space. CVPR, 2017.
> > >
> > > [50] Weili Nie, Arash Vahdat, and Anima Anandkumar. Controllable and compositional generation with latent-space energy-based models. NeurIPS, 2021.

---

### Official Review · Reviewer_gbfq · 2022-07-11

**Rating:** 6
**Confidence:** 5
**Soundness:** 3 good
**Presentation:** 3 good
**Contribution:** 2 fair

**Summary:**

This paper proposes PromptGen, a method for sampling pre-trained generator networks (e.g. StyleGAN) conditioned on a prompt (label, text, etc) using off-the-shelf discriminative models (classifiers, CLIP etc). The proposed method uses an invertible neural network (INN) to express a distribution p(z ; C) over generator latents z given constraint C. A prompted sample is obtained by sampling p(z ; C) and then passing through the generator to obtain a data-space sample (all experiments are conducted with images but in principle the method could be applied to other modalities). INNs are trained either for a specific instant of the prompt C, or in a prompt conditional setting.

The method is demonstrated on a range of tasks, including de-biasing StyleGAN, and generating conditioned on a text prompt (with CLIP).





**Questions:**

* The prospect of using such models to reveal the limitations of pre-trained models (e.g. CLIP and the makeup example) is potentially very useful. But how do we know if the limitation belongs to the prompting model, or if it is a limitation of the INR-based p(z ; C)? For some prompts, p(z ; C) might be very complex, and therefore challenging to learn. So how do we know that we haven't just failed to (fully) learn the mapping when training the INR? If there isn't a good way of knowing, then this is a limitation, as in general it won't be possible to confidently use the proposed method to diagnose other models.
* For the text conditional experiments, is the INN trained just for a single prompt? Why not train a text-conditional model? It is impractical to train separate models for each text input, and is a limitation if it is not possible with the proposed approach.
* Can PromptGen be applied to diffusion models?
* Could you expand on this comment in the limitations:  "Except for the case of generative models with a class-embedding space, PromptGen focuses on mode-seeking and mode-reweighting instead of domain adaptation" - I didn't really understand what was meant by domain adaptation vs mode seeking in this context.

**Limitations:**

* The authors adequately addressed the fact that the method is dependent on the data coverage of the generator network used. E.g. if you're using StyleGAN you're going to get faces and not spaceships.
* The need to train a potentially very powerful conditional INN for complex tasks such as text-conditional generation wasn't really addressed (as mentioned in the weaknesses section). For this use case the burden of training such a network could easily be so high that a practicioner may prefer an iterative inference time method (like PPGN), even with its higher inference costs.
* Potential negative societal impacts were not really discussed, however the experimental work on de-biasing pre-trained models (StyleGAN, CLIP) communicates to the reader that bias is an issue with such models that should be addressed.

**Strengths And Weaknesses:**

Strengths
* Unlike iterative inference time methods like PPGM, PromptGen can be applied (after training) to sample latents in a single forward pass, resulting in significantly improved inference speeds. This is particularly useful in cases where the INN can effectively represent a constraint that can be applied to all samples, e.g. StyleGAN race de-biasing.
* The compositionality of the proposed method, where prompting mechanisms can be applied in sequence, is a desirable property.
* The experiments: race / gender de-biasing, text-conditioning are well chosen for their real-world importance.

Weaknesses
* The broad usefulness of the method depends on the quality and coverage of the pre-trained generator network. GAN generators in general are prone to mode-dropping, especially those trained on complex natural image distributions (BigGAN). This is addressed in the limitations section.
* For complex conditional tasks, like text-conditioned generation, the INN will have to learn a very complex mapping (e.g. from text -> latents). This will require lots of data / compute / optimization etc and as such might be prohibitive in practice.
* In general there is a balance between the benefits of amortization: fast inference speeds, and the costs: training time and compute. And which is preferable depends on a user's constraints. I think the paper would benefit from a discussion of these issues.

---

> ### Author Response · Authors · 2022-08-01
> **Author response**
>
> **Quality and coverage of pre-trained generators are a bottleneck**
>
> We agree. The main focus of this work is to sample from a specific output region (or to reweight the output distribution for de-biasing) of a pre-trained network. This limitation is also related to the last of your questions (mode-seeking and domain adaptation).
>
>
> **Trade-off between inference-time optimization and amortization (training-time optimization)**
>
> Thanks for pointing it out! Amortization is helpful when one wants to reuse a controlled distribution many times, which we believe is the case for (1) creating controllable training data (for recognition tasks) from generative models, (2) debiasing generative models, and (3) pose-conditioned face modeling. Moreover, we recall that training the INN for text-control experiments only takes hundreds of steps to converge. In contrast, inference-time optimization methods (e.g., the MCMC-based PPGM) take 50 optimization steps to generate one sample. In the updated version, we discuss this issue in Section 4.6.
>
>
> **Approximation error of PromptGen should be considered in the error breakdown.**
>
> We agree. This approximation error can be investigated by comparing PromptGen with MCMC-based PPGM, whose error diminishes given enough optimization steps. In Section 4.6 (Section 4.5 in the original version), we reported a quantitative comparison between PromptGen and PPGM, which shows that PromptGen and PPGM have similar controllability (when the control is successful). To verify the failure cases also happen to PPGM, we ran an additional experiment of “photo of a man without beard” for PPGM. Results are reported in Figure 20 in the updated version.
>
>
> **Why not train a text-conditioned model for many possible text descriptions?**
>
> One can definitely train a text-conditioned model given a list of text descriptions, but it can limit the generalization to novel text descriptions. One related experiment is the pose-conditioned experiment, in which PromptGen is conditioned on the pose parameter; in this experiment, generalization is enabled by conditioning since we can easily sample all possible pose parameters from its domain during training.
>
>
> **Applicability of PromptGen to diffusion models**
>
> Thanks for pointing it out! We added one experiment on Diffusion Autoencoder in Figure 5(d). It is a hybrid model of diffusion models (specifically DDIM) and autoencoders. However, we did not run an experiment on the original DDPM since it does not have a typical latent code.
>
>
> **Mode-seeking v.s. domain adaptation**
>
> *Mode-seeking* means *controllability* in this paper, a term sometimes used in literature to refer to sampling from a particular mode learned by the generative model. *Domain adaptation* means to finetune the generative model to generate samples from a domain not seen during training. For example, StyleGAN-NADA, a domain adaptation method, can finetune a StyleGAN trained on FFHQ (real faces) to generate Pixar-like faces.
>
> The main difference between *mode-seeking* and *domain adaptation* is the support set of the output distribution. In *mode-seeking*, the support set is the same as (or is a subset of) the original distribution’s support set. In contrast, in *domain adaptation* the support set can be drastically different from the original distribution’s support set. In Figure 4(b), we showed that *domain adaptation* fails when *mode-seeking* should be used (i.e., the set of baby faces is a subset of the set of human faces). On the other hand, *mode-seeking* can also fail when we want to generate something that the generative model never sees (e.g., Pixar-like faces). In some cases, maybe we need both *mode-seeking* and *domain adaptation* (e.g., Pixar-like baby faces). PromptGen allows for this combination: the functional composition $G \circ f_\theta$ is a mapping from $\mathbb{R}^{d}$ to $\mathcal{X}$, i.e., a generative model (we refer to **Clarification: energy composition v.s. functional composition** section in **General response to all reviewers**); therefore, we can directly apply StyleGAN-NADA to $G \circ f_\theta$.  We will clarify this in the final version of the paper.

---

> > ### Comment · Reviewer_gbfq · 2022-08-09
> > **Thank you for the response, but some issues remain**
> >
> > Thank you for your detailed response.
> >
> > > This approximation error can be investigated by comparing PromptGen with MCMC-based PPGM, whose error diminishes given enough optimization steps. In Section 4.6 (Section 4.5 in the original version), we reported a quantitative comparison between PromptGen and PPGM, which shows that PromptGen and PPGM have similar controllability (when the control is successful). To verify the failure cases also happen to PPGM, we ran an additional experiment of “photo of a man without beard” for PPGM. Results are reported in Figure 20 in the updated version.
> >
> > This is an empirical argument: in the cases you tried, PromptGen achieves similar controllability to PPGM. My original question was whether PromptGen can be relied on to diagnose limitations of another model (e.g. CLIP). This experiment supports that claim, but can we be sure that we will achieve low approximation error in general, as we can with PPGM?
> >
> > > One can definitely train a text-conditioned model given a list of text descriptions, but it can limit the generalization to novel text descriptions
> >
> > Why would this be the case? After all, there now exist very good generative models of images given text.

---

> > > ### Author Response · Authors · 2022-08-09
> > > **Author response**
> > >
> > > **Approximation error**
> > >
> > > Thanks for pointing this out! The approximate error can be measured by $D_{\text{KL}}(p_{\theta}(\boldsymbol{z}) || p(\boldsymbol{z} | \mathcal{C}))$. This KL divergence is defined in Eq. (9). However, it is worth noting that $\log Z$ is expensive to estimate in practice (recall that this partition $\log Z$ does not depend on $\theta$ so we can safely discard it in the training objective). We will discuss this approximation error in more detail in the final version.
> > >
> > > **Text-conditioned models**
> > >
> > > We should definitely elaborate on this!
> > >
> > > > One can definitely train a text-conditioned model given a list of text descriptions, but it can limit the generalization to novel text descriptions.
> > >
> > > By “text-conditioned model” we mean a text-conditioned PromptGen. To train it, one needs a list of domain-specific (faces, buildings, etc.) descriptions; however, given a description not seen during training, this text-conditioned PromptGen may fail to generalize.
> > >
> > > > After all, there now exist very good generative models of images given text.
> > >
> > > By “very good generative models of images given text” we believe you mean models like DALL-E 2, Imagen, and Parti. Our observation working with some of these models is that the quality of these models in highly specialized domains (e.g., faces) still lags behind domain experts such as StyleGAN. For example, one may try the following prompts using DALL-E 2 / DALLE-mini API: “a photo of a baby’s face” and “a photo of an Asian female”.

---

### Official Review · Reviewer_byc9 · 2022-07-11

**Rating:** 6
**Confidence:** 3
**Soundness:** 3 good
**Presentation:** 2 fair
**Contribution:** 3 good

**Summary:**

This paper introduces an approach to use supervision from off-the-shelf pre-trained models, including classifiers, scoring functions, and multi-modal encoders, as controls for the generation of images with custom properties. In particular, the proposed approach transforms random noise, using an invertible neural network, that is fed to a pre-trained generative model.

**Questions:**

* The authors attempt at differentiating this paper from prior work suggests that [prior methods] “are either model-dependent (i.e., requiring a well-structured style space) or label-intensive (i.e., requiring all training samples to be labeled for explicit conditions), limiting their generality and practical use.” However, there are several prior works that enable sampling from a specific region of the latent space in a frozen generative model without doing any optimization at inference time (e.g. [1]) (not needing labels at training time, using a frozen generative model, using interpolation of previously learnt vectors at inference time, applicable to both GAN and VAE models). Additionally, it is unclear what being “model-dependent” means here. Does it mean learning to navigate the latent space that is specific to a single pre-trained generative model and may not generalize to other pretrained generative models? If so, the parameters $\theta$ in PromptGen also rely on the pretrained generative model at hand (In Algorithm 2, the gradients pass through the frozen $G_{\phi}$.) Conceptually, both PromptGen and the aforementioned approaches apply to different pre-trained generative models, though the parameters are going to be tuned for each frozen model. The authors might have missed such prior research in their literature review or the writeup does not properly communicate that. Please clarify the distinctions and positioning of this work.

*  Method clarification: It is unclear where the different formulations of energy for the classifier/CLIP type models/and scoring functions (i.e. the inverse graphics case) come from. Is this well-defined in an EBM? Is this inspired by prior work? Is this a design choice?

*  Prompt tuning uses a (small) dataset for tuning additional parameters that are (usually) prepended to a meaningful (textual) input to steer the model’s outputs in a direction of interest. Some variants do it in the discrete natural language format and some in continuous embedding space. Some form of concatenation to an existing meaningful input is the fingerprint of what is usually dubbed as “prompt tuning”. So in my opinion, drawing a parallel between “learning transformations over random noise” and “prompt tuning” seems a bit tangential and potentially distracting at first. Perhaps authors meant other connections? If yes, would appreciate clarifications.

[1] Shen, Yujun, et al. "Interpreting the latent space of gans for semantic face editing." Proceedings of the IEEE/CVF conference on computer vision and pattern recognition. 2020.


**Limitations:**

The authors have briefly described the societal impact of their work. Additionally, one can argue that more granular controls for generative models might make it easier for generating targeted Deepfakes and its resulting potential negative impacts.

**Strengths And Weaknesses:**

+This paper includes a through set of experiments that showcase the effectiveness of the proposed approach across several pre-trained generative models (e.g. StyleNeRF, NVAE, BigGAN, StyleGAN2) and different types of controls (binary properties, continuous values, text descriptions).

+A unified approach to custom controls for successfully steering pre-trained generative models can facilitate several down-stream applications and has high significance. In particular, success in compositional controls makes this work even more interesting.

-Both the positioning of the work and the explanation of the method can be further clarified.  See below for more details.

---

> ### Author Response · Authors · 2022-08-01
> **Author response**
>
> **What does model-dependent mean? Distinctions from previous approaches on latent (or style) code editing/interpolation**
>
> We agree that “model-dependent” needs further clarification. We described previous methods on latent (or style) code editing/interpolation as “model-dependent” because they put *non-trivial assumptions of locality and interpolation* on the latent (or style) space. Specifically, latent (or style) code editing assumes that image semantics can be guided by locally modifying the latent (or style) code of each image; latent (or style) code interpolation (e.g., the Slerp interpolation) assumes that any point on the interpolation of two latent (or style) codes corresponds to an image whose semantics is also interpolated. These assumptions do not always hold for every control. As we showed in Figure 4(a), the local editing-based StyleCLIP cannot model “a photo of a baby” well, and we attributed it to the fact that not all images’ latent code can be locally edited into a baby. Moreover, since PromptGen’s $f_\theta$ is an invertible mapping from $\mathbb{R}^{d}$ to $\mathbb{R}^{d}$, local editing is a special case of $f_\theta$.
>
> In the original version, we cited works on latent (or style) code editing in Line 62: “local editing of the learned representation, e.g, “style” codes [1, 54, 44, 64, 74]”. We are updating the above arguments in Sections 1, 2, and 4.
>
> Finally, given one specific control, one never knows which method works the best. We hope our method will serve as an effective and efficient tool in downstream applications.
>
>
> **Where do the energy functions come from?**
>
> The classifier energy is based on the Bayes rule and temperature-adjusted distributions, commonly used by previous works that we cannot find an appropriate reference. The CLIP energy (especially the differentiable augmentation part) is inspired based on FuseDream [44], which we cited properly, while they did not use it for energy-based modeling. We designed the inverse graphics energy, whose interpretation (e.g., what kind of distribution it models) is provided in Appendix B.1. We note that all energy functions are not unique and can take other forms, and should be flexibly adjusted based on the application. We will further clarify this in the final version.
>
>
> **Connection to prompt tuning**
>
> Thanks for pointing this out! The major connection is that PromptGen learns a distribution over a pre-trained generative model’s input space, so arbitrary controls can be achieved without finetuning the pre-trained model. We agree that “Prompt Generation” is ambiguous, which sounds like a task that aims at generating text prompts.

---

### Official Review · Reviewer_5eQN · 2022-07-13

**Rating:** 7
**Confidence:** 3
**Soundness:** 2 fair
**Presentation:** 3 good
**Contribution:** 3 good

**Summary:**

This work proposes a method to control pre-trained generative models (e.g. to condition samples on a text prompt, or to control the value of an attribute, or to debias samples). The control can be specified using a different model (e.g. CLIP or inverse graphics or a classifier) via an energy-based formulation that outputs a controlled-version of the pre-trained generative model.

**Questions:**

- Functional composition is an important property that needs further evidence. Consider providing results that show how well CLIP captions can be composed (e.g. C_1=“photo of an Asian man” and C_2=“photo of a person with eyeglasses” and C_3=“photo of a bald person”). If you could expound on the performance difference between using a multi-component control C (as suggested in Figure 2’s caption) or using two separate controls C_1 and C_2, that would be super helpful.
- Table 2: could you please add at least one more column here so we can ensure the attributes aren’t cherry-picked? How about hairColor (say “black” if you need a binary attribute)?
- Fig 9: the latents learnt by the GAN looks fairly well separated between the two clusters. How well is PromptGen able to cope with less structured latent spaces?
- Could you clarify whether equation 8 holds for all invertible NNs, or only the architecture you've chosen?

**Limitations:**

Section F in the appendix covers some interesting failure modes. But it places the burden of failure mostly on the pre-trained generative model or control model. It would be nice to see where the overall idea/method might fail.

**Strengths And Weaknesses:**

**Strengths**
- Compatibility with a wide range of control models (classifiers, inverse graphics, embedding)
- Compatibility with a wide range of generative models (results include StyleNerf, NVAE, StyleGAN2, and BigGan)
- Compositionality of controls
- No optimization at inference
- The writing is easy to follow. (But here's a specific point. The following motif is repeated at several points in the paper: PromptGen leverages “the knowledge of various off-the-shelf models” for distributional control of pre-trained generative models. It is quite unclear especially in the abstract what “off-the-shelf models” you are referring to, and whether you are still talking about generative models. Perhaps it would be clearer to state that you are trying to control pre-trained generative models using knowledge from “other models” rather than “off-the-shelf models”?)

**Weaknesses/concerns**:
- Most results are on faces. The authors could have evaluated on different data or chosen a wider variety of prompts. So far it is only clear that the model can control/yield pictures of babies, cats, and persons (especially controlling for age, make-up, or race).
- When evaluated on non-facial data, the results lack diversity (PromptGen on ImageNet 512 in Figure 5(d)). “A photo of a glow and light dog” seems to yield a single specimen. The melancholy robot always has the same helmet and the background is a consistent color. There’s no further results from ImageNet in the supplementary. Also the size of the generated samples is too small given they were 512x512.
- CLIP control doesn’t always seem to work well e.g. in Fig 12(a), at least a third of the images have cats with open eyes. It is unclear whether it is CLIP to blame, or a bias in the image dataset.

---

> ### Author Response · Authors · 2022-08-01
> **Author response**
>
> **Most results are on faces**
>
> We used faces (real faces or faces in Met Art collections) as the main data for two reasons (1) humans are more sensitive to artifacts in the generated faces (than generated cars, cats, churches, etc.), and (2) there are well-studied/used face repositories that are good for comparison with previous approaches. In addition to faces, in Figure 5(c), Figure 12 (Appendix), and Figure 13 (Appendix), we provided results on more datasets such as Landscape, LSUN Church, and AFHQ Cats. In appendix E, we also provided some preliminary results on 3D human faces.
>
>
> **Visualizations are too small for high-res images**
>
> Please, zoom in within the PDF for better visualization (although they are already downsampled). Since we want to keep the PDF reasonable in size, we will include full-res samples (mostly 1024 x 1024 and 512 x 512) on our accompanying website. We also had some high-res samples in Appendix D and F.
>
>
> **Lack of diversity on ImageNet**
>
> First, we refer to **Clarification: PromptGen in the class-embedding space** section in **General response to all reviewers**. The ImageNet experiment is mainly designed to show that PromptGen can not only model distributions in the $\boldsymbol{z}$-space, but also the class-embedding space, an extension of our main method. However, we agree that this extension suffers from low diversity; in the updated version, we mentioned this limitation. Thanks for pointing this out.
>
>
> **How to do error breakdown in cases where CLIP control doesn’t work well?**
>
> That is a good question. We provided a preliminary answer in Appendix F. Specifically, we used CLIP to retrieve images from a large set of 400M images; if the retrieved images are faithful to the text description, then CLIP should not be blamed. Using this idea, we showed in Appendix F that CLIP could not model “without beard” and is gender-biased when modeling “a person without makeup”.
>
> Regarding your question on “a cat with closed eyes”, we find that almost all retrieved cats from the 400M images have closed eyes; therefore, the failure probably comes from the low density in the training data. Moreover, the energy-based model reweighs the distribution instead of enforcing constraints, but picking useful images from the reweighted distribution will be more efficient than picking from the original one. We will describe our findings in this regard in the final version of the paper.
>
>
> **More analysis on functional composition**
>
> First, we refer to **Clarification: energy composition v.s. functional composition** section in **General response to all reviewers**. Decomposing a complex description into two simpler ones is interesting and falls into what we mean by “energy composition”. We added a new section (Section 4.5) to discuss this issue. Specifically, we show that control with a complex sentence (i.e.,*a photo of a bald black man with beard*) is not very successful. We then decompose the complex sentence into three simpler ones (i.e., *a photo of a black man*, *a photo of a bald man*, and *a photo of a man with beard*), and we find that by tuning the weight $\lambda_i$ for each of them (e.g., we need larger $\lambda_i$ for *a photo of a black man*), we have better control performance.
>
>
> **Additional experiments for Table 2**
>
> In the updated version, we reported the hair color de-biasing results in Table 2, and PromptGen achieves nearly perfect de-biasing performance (we will also report baseline results if time permits). Moreover, all attributes and attribute combinations reported in Table 2 come from Table 1 of FairStyle [30].
>
>
> **Latent vectors learned by GAN look fairly well separated on synthetic data (Fig. 9)**
>
> The motivation of this synthetic experiment is to provide visual intuition about what PromptGen learns. We agree that this is not representative of real data, and for this reason, we put it in the appendix. The paper provides various real data experiments to show PromptGen’s ability to cope with less structured latent spaces.
>
>
> **Does Eq. (8) hold for all INNs?**
>
> This is correct; Eq. (8) holds for all INNs.

---

### Author Response · Authors · 2022-07-28
**General response to all reviewers**

Thank you for your valuable feedback! Here, we would like to address some questions raised by more than one reviewer.

**Clarification: energy composition v.s. functional composition**

By “energy composition” we mean $E_{\mathcal{C}}(\boldsymbol{x}) = \sum_{i=1}^{M} \lambda_i E_i(\boldsymbol{x}, \boldsymbol{y}_i)$ in Eq. (1) and Eq. (2), where the control $\mathcal{C}$ is composed of $M$ independent properties $\{\boldsymbol{y}_1, \ldots, \boldsymbol{y}_M\}$, e.g., $\boldsymbol{y}_1$ can be a text description and $\boldsymbol{y}_2$ can be an attribute. This energy composition is useful when multiple controls **can** be specified simultaneously. In the updated version, we also added a section (Section 4.5) to discuss how energy composition helps us decompose complex controls into simpler ones.

By “functional composition” we mean the iterative control described in Figure 2, where we treat a PromptGen-controlled generative model $G \circ f_\theta$ as a new generative model. This is based on the fact that $f_\theta$ is a mapping from $\mathbb{R}^{d}$ to $\mathbb{R}^{d}$ and $G$ is a mapping from $\mathbb{R}^{d}$ to $\mathcal{X}$; therefore $G \circ f_\theta$ is a mapping from $\mathbb{R}^{d}$ to $\mathcal{X}$, i.e., a generative model. Functional composition (or iterative control) is useful when multiple controls **cannot** be specified simultaneously. For example, in Section 4.4, we provided a case where we want to generate a gender-debiased image distribution over people without makeup. In this case, we need to *first* learn the distribution over people without makeup and *then* de-bias this distribution, since debiasing a distribution requires knowing what the distribution looks like.

Thanks for pointing this out. In the updated version, we experiment with both energy composition and functional composition in Section 4.5 and Section 4.6. We will further clarify these terms in the final version of the paper.

**Clarification: PromptGen in the class-embedding space**

We would like to distinguish two settings when using a class-conditional GAN as the generative model: (1) PromptGen in the $\boldsymbol{z}$-space of a class-conditional GAN and (2) PromptGen in the class-embedding space of a class-conditional GAN. The first setting is useful for finding modes in a particular class, while the second helps generate novel objects that are not one of the 1000 classes in ImageNet.

*PromptGen in the $\boldsymbol{z}$-space of a class-conditional GAN*: one first specifies a class $c$ in ImageNet (e.g., $c=$ cat) and one control (e.g., “a photo of a cat sitting on a mat”), and PromptGen learns a distribution in the $\boldsymbol{z}$-space. Under this setting, the goal of PromptGen is to find a distribution over latent codes $\boldsymbol{z}$ such that $G(z, \text{Emb}(c))$’s are “sitting cats”. This setting is the same as the unconditional setting discussed throughout this paper, given the fact that $G(\cdot, \text{Emb}(c))$ has the same form as an unconditional generative model.

*PromptGen in the class-embedding space of a class-conditional GAN*: one specifies an out-of-domain control (e.g., “a photo of a glow and light dog”), and PromptGen learns a distribution in the class-embedding space. This setting exploits the observation that a class-conditional GAN can generate out-of-domain samples when the provided class embedding is not one of the classes in ImageNet. Under this setting, PromptGen aims to find a distribution over the class embeddings $y$ such that $G(z, y)$’s are “photos of a glow and light dog”. However, results in Figure 5(e) [previously 5(d)] show that the diversity of images sampled from such control is limited; in the updated version, we mentioned this limitation.

---

### Author Response · Authors · 2022-08-02
**Societal impact statement**

We would like to thank the reviewers for pointing out the issue of societal impacts. In the updated version, we included the following statement in Appendix G.

With the improvements in generative models for images, DeepFake technology has become more accessible. Like any new technology, it is a double-edged sword, and
it is crucial to research and comprehend the possible advantages and disadvantages of generative models for society.

On the positive side, we show that PromptGen can be used to de-bias pre-trained generative models and to reveal biases learned by text-image models (e.g., CLIP), indicating that PromptGen might be a useful tool for fair AI if used appropriately. The efficient inference provided by PromptGen also helps reduce the computational expense, which has a positive impact on the environment. Better controllability, however, unavoidably makes it simpler to synthesize targeted pictures, which might have detrimental social effects in creating deceptive media  (e.g., DeepFakes) or privacy leaks (e.g., identity-conditioned human face synthesis). To battle these cases, there are current technologies that can detect fake media effectively, and we expect companies and users to use these technologies to distinguish what is real from fake. We encourage practitioners to consider these risks when using PromptGen to develop systems.

---

### Meta-Review · Area_Chair_ycRo · 2022-08-30

**Recommendation:** Accept
**Confidence:** Less certain

**Metareview:**

This work concerns a unifying method for repurposing "off the shelf" conditional models in order to define an energy-based model of vectors in the latent space of a pre-trained generative model, for the purpose of controlling synthesis, and a feed-forward approximation using invertible neural networks. The authors present several use cases and experiments on each across a range of different model types.

Reviewers were positive on the presentation, originality and usefulness, and generally felt the experiments were well chosen. There were some concerns regarding discussion of societal impact (gbfq), the fact that most results involved faces and those that didn't were less compelling (5eQN), and clarity around the derived energy function and positioning relative to prior work (byc9). Most concerns were addressed in rebuttal, however QXTM felt quantitative results evaluating controllability, specifically, left much to be desired, and lowered their score following a rebuttal that they felt failed to address this issue.

Based upon the discussion and my own reading of the paper, the AC views this work in an overall positive light, the valid concerns of QXTM notwithstanding. With some reservations, I recommend acceptance.

**Award:**

No

---

### Decision · Program_Chairs · 2022-09-14

Accept